# Block Coordinate Descent for Neural Networks Provably Finds Global Minima

**Shunta Akiyama**
CyberAgent
akiyama_shunta@cyberagent.co.jp

## Abstract

In this paper, we consider a block coordinate descent (BCD) algorithm for training deep neural networks and provide a new global convergence guarantee under strictly monotonically increasing activation functions. While existing works demonstrate convergence to stationary points for BCD in neural networks, our contribution is the first to prove convergence to global minima, ensuring arbitrarily small loss. We show that the loss with respect to the output layer decreases exponentially while the loss with respect to the hidden layers remains well-controlled. Additionally, we derive generalization bounds using the Rademacher complexity framework, demonstrating that BCD not only achieves strong optimization guarantees but also provides favorable generalization performance. Moreover, we propose a modified BCD algorithm with skip connections and non-negative projection, extending our convergence guarantees to ReLU activation, which are not strictly monotonic. Empirical experiments confirm our theoretical findings, showing that the BCD algorithm achieves a small loss for strictly monotonic and ReLU activations.

## 1 Introduction

Deep learning has driven remarkable progress across a wide range of fields, including computer vision, natural language processing, and reinforcement learning, achieving state-of-the-art results on numerous tasks. Despite these empirical successes, the theoretical understanding of the training dynamics and optimization behavior of deep neural networks remains elusive, primarily due to the highly non-convex structure of their loss landscapes [20]. A central open problem is the establishment of convergence guarantees to global minima for gradient descent algorithms, particularly those implemented through backpropagation in deep architectures comprising multiple layers. The neural tangent kernel (NTK) framework [17] has provided partial theoretical insights by approximating the training dynamics by a linearized one within a reproducing kernel Hilbert space (RKHS). However, this linearized perspective does not fully capture the empirical efficacy of deep learning models. In practice, deep learning often surpasses the performance of kernel methods, including those based on NTK, suggesting that the NTK regime captures only a limited aspect of optimization capabilities.

As an alternative to backpropagation-based gradient methods, block coordinate descent (BCD), a framework rooted in mathematical optimization (see, e.g., [34]), optimizes partitioned variable blocks iteratively while holding others fixed, offering computational efficiency through partial parameter updates. Its structure also supports parallel and distributed implementations [10, 21], making it well-suited for large-scale neural network training.

Given the highly non-convex nature of neural network loss landscapes, BCD-based approaches have emerged as promising alternatives to conventional gradient-based approaches [11, 6, 19, 43, 29, 41, 24, 31, 42, 38]. A widely adopted strategy in this setting is to partition the network parameters by layer, treating the weights of each layer as individual blocks, allowing for sequential or alternating updates, as illustrated in Figure 1. This layer-wise decomposition enables the reformulation of the

39th Conference on Neural Information Processing Systems (NeurIPS 2025).

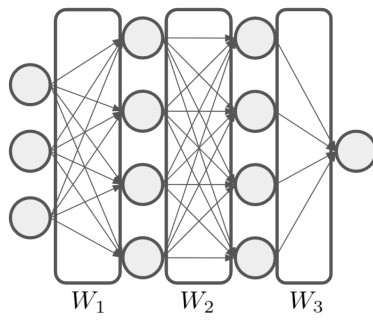 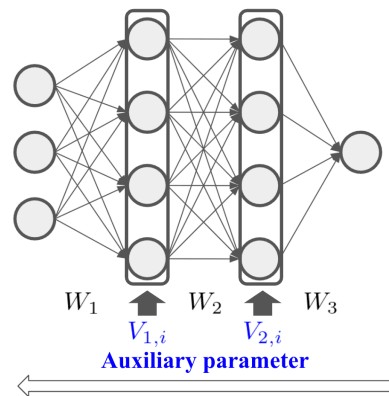

**Update by backpropagation**

$W_1$ $W_2$ $W_3$

$V_{1,i}$ $V_{2,i}$
**Auxiliary parameter**

**Update by block coordinate descent**

Figure 1: Graphical comparison between backpropagation and block coordinate descent. In contrast, the block coordinate descent approach introduces auxiliary variables $V_{j,i}$, which serve as approximations of the hidden layer outputs, enabling layer-wise updates and a decoupled optimization structure (see Section 4 for details).

original global loss into a series of sub-objectives, each localized to a specific layer. These sub-problems typically exhibit more tractable optimization properties compared to the full loss function, thereby facilitating more efficient and stable training dynamics.

Building on the practical advantages of block coordinate descent (BCD) in neural network training, recent research has increasingly examined its theoretical properties, particularly its convergence behavior. However, the current literature on BCD for neural networks [43, 41, 42, 38] has been limited to establishing convergence to stationary points, that is, points where the gradient vanishes. While such results are of theoretical interest, they do not ensure convergence to global optima, particularly in light of the highly non-convex loss landscapes intrinsic to deep neural networks [20, 32].

Understanding how neural network training can achieve global minima has emerged as a central problem in the theoretical study of deep learning. Although BCD offers a promising alternative to gradient-based methods, existing guarantees have not addressed this critical aspect, remaining confined to local convergence results. To address this gap, we aim to establish convergence guarantees to global minima for BCD applied to neural network training. Specifically, we consider multi-layer neural networks and analyze a BCD-type algorithm in which each block update is performed via standard (vanilla) gradient descent. Our main contributions are summarized as follows:

- We establish a global convergence guarantee for a block coordinate descent (BCD) algorithm applied to the training of deep neural networks with strictly monotonically increasing activation functions. To the best of our knowledge, this is the first theoretical result that ensures convergence to global minima in deep neural networks with an arbitrary number of layers, without relying on assumptions associated with the neural tangent kernel (NTK) regime.

- Under the assumption that the training data are i.i.d., we derive generalization bounds for networks trained via BCD. A cornerstone of our analysis is the boundedness of the weight matrices at each layer—a property we establish as a direct consequence of the convergence proof. Leveraging this boundedness together with the Rademacher complexity framework introduced by [8], we derive upper bounds on the generalization gap.

- Since ReLU does not satisfy the monotonicity condition required by our initial analysis, the corresponding convergence results are not directly applicable. To address this issue, we propose a modified BCD algorithm that integrates skip connections [16] and non-negative projection steps. This augmentation enables us to extend our global convergence guarantees to networks employing ReLU activations, thereby broadening the practical relevance and applicability of our theoretical contributions.

## 2  Related Work

**Convergence Guarantees for Gradient Descent and Stochastic Gradient Descent in Neural Networks**   The convergence properties of gradient descent (GD) and stochastic gradient descent (SGD) in the context of neural network training have been the subject of extensive theoretical investigation in recent years. A prominent line of research focuses on the *neural tangent kernel* (NTK) regime [17, 3, 4, 13, 44], wherein the training dynamics of highly overparameterized neural networks are approximated by linear models in a reproducing kernel Hilbert space (RKHS). While this framework facilitates global convergence analysis via convexity, it fails to capture the feature learning capability of neural networks. In particular, models trained under the NTK regime often exhibit minimal parameter movement from their initialization, thereby behaving more like kernel methods than adaptive learners. In contrast, our analysis operates outside of this kernel-based setting and does not rely on such overparameterization assumptions to ensure global convergence.

An alternative theoretical framework is the *mean-field* (MF) regime [28, 12, 22, 35, 30, 27], which interprets training dynamics as the evolution of probability measures over the parameter space. This formulation enables the conversion of the original non-convex optimization problem into a convex optimization over probability distributions. While the MF regime avoids the limitations of the NTK approach by preserving feature learning behavior and allowing global convergence analysis, most existing results are restricted to shallow architectures, typically two or three layers. In contrast, our work provides global convergence guarantees for neural networks with an arbitrary number of hidden layers.

More recently, [7] introduced the concept of restricted strong convexity (RSC) to analyze neural network training. This framework derives global convergence by assuming a correlation between the gradients and outputs of neural networks throughout training. However, the validity and general applicability of this correlation assumption remain to be fully established.

**Generalization Error Bounds for Multi-Layer Neural Networks**   The study of generalization properties in deep neural networks has also seen considerable progress [26, 37, 8, 25, 15, 9, 5, 33]. These works develop upper bounds on generalization error by evaluating the capacity of neural networks using various complexity measures, such as VC-dimension, norms of weight matrices, and spectral properties. However, most of these studies focus solely on capacity control and do not address the optimization dynamics during training. While some recent studies have extended generalization analysis to networks with more than two layers with global convergence guarantees, many remain constrained to three-layer architectures [1, 2].

## 3  Preliminaries

### 3.1  Notations

For $x \in \mathbb{R}^d$, $\|x\|$ denotes its Euclidean norm. We denote the $d$-dimensional identity matrix by $I_d$. For $A \in \mathbb{R}^{n \times m}$, the Frobenius norm is defined as $\|A\|_F := \sqrt{\sum_{i,j} A_{ij}^2}$, and the operator norm is defined as $\|A\|_{op} := \sup_{\|x\| \leq 1} \|Ax\|$. For two symmetric matrices $A$ and $B$, we write $A \prec B\,(A \preceq B)$ if and only if the matrix $B - A$ is positive (non-negative) definite. For $x = (x_1, \ldots, x_d)^\top \in \mathbb{R}^d$, $\mathrm{diag}(x) \in \mathbb{R}^{d \times d}$ denotes a diagonal matrix whose $j$-th diagonal entry is $x_j$. We denote $\min\{a, b\} =: a \wedge b$.

### 3.2  Problem Settings

We consider a supervised learning setup where we are given $n$ training examples $\mathcal{D} = \{(x_i, y_i)\}_{i=1}^n$, with input vectors $x_i \in \mathbb{R}^{d_{\mathrm{in}}}$ and corresponding labels $y_i \in \mathbb{R}^{d_{\mathrm{out}}}$. We define the data matrix as $X = (x_1, \ldots, x_n)^\top \in \mathbb{R}^{n \times d_{\mathrm{in}}}$. Throughout this work, we consider a high-dimensional regime, where the data points do not exceed the input dimension, i.e., $n \leq d_{\mathrm{in}}$. To ensure the well-posedness of our theoretical results, we impose the following assumption on the data matrix:

**Assumption 3.1** (Full-rank data matrix)**.** The matrix $X$ has full row rank, i.e., $\mathrm{rank}(X) = n$.

This assumption is essential for establishing global convergence guarantees. As demonstrated in the proof of our main result, the existence of global minima cannot be ensured without Assumption 3.1.

We consider a fully connected feedforward neural network with $L$ layers, defined by
$$f_{\mathrm{NN}}(x) \coloneqq W_L \sigma(W_{L-1}\sigma\left(\ldots \sigma\left(W_2\sigma(W_1 x)\right)\ldots\right)),$$
where $\sigma$ is an element-wise activation function. The weight matrices are specified as $W_1 \in \mathbb{R}^{r \times d_{\mathrm{in}}}$, $W_j \in \mathbb{R}^{r \times r}$ for $j \in \{2, \ldots, L-1\}$, and $W_L \in \mathbb{R}^{d_{\mathrm{out}} \times r}$. We assume all hidden layers share a common width $r$.

Next, we introduce our assumption on the activation function.

**Assumption 3.2** (Activation). $\sigma : \mathbb{R} \to \mathbb{R}$ is monotonically increasing and satisfies $\sigma(0) = 0$. Especially, there exists a constant $0 < \alpha < 2$ such that $\inf_{x \in \mathbb{R}} \sigma'(x) \geq \alpha$ holds[1]. Moreover, $\sigma$ is $\ell$-Lipschitz, i.e., for any $u_1, u_2 \in \mathbb{R}$, $|\sigma(u_1) - \sigma(u_2)| \leq \ell|u_1 - u_2|$ holds.

A typical example of an activation function satisfying Assumption 3.2 is the LeakyReLU activation defined by $x \mapsto \max\{x, ax\}$ for $a < 1$, which satisfies the assumption with $\alpha = a$ and $\ell = 1$. On the other hand, the standard ReLU activation $x \mapsto \max\{x, 0\}$ does not satisfy Assumption 3.2. Nevertheless, we provide a global convergence result for networks with ReLU activation through a modified BCD algorithm, as detailed in Section 6.

Given this neural network formulation, we formalize the supervised regression problem as
$$\min_{\mathbf{W}} \sum_{i=1}^{n} \|f_{\mathrm{NN}}(x_i) - y_i\|^2, \tag{1}$$
where $\mathbf{W} = (W_1, \ldots, W_L)$ denotes the collection of weight matrices. One of the most commonly used methods for solving (1) is the (stochastic) gradient method, which updates parameters based on the gradient of the loss function. In contrast, we employ a layer-wise optimization method known as *block coordinate descent*, which we introduce in the following section.

## 4  Block Coordinate Descent

In this section, we first introduce the basic concept of *block coordinate descent* (BCD) and then describe the specific BCD algorithm considered in this paper. Originally developed within the field of mathematical optimization (see, e.g., [34]), BCD is a general framework for solving high-dimensional optimization problems by partitioning the set of variables into disjoint blocks and optimizing each block iteratively while keeping the others fixed.

In our setting, instead of directly minimizing the original loss in (1), we introduce auxiliary variables $V_{1,i}, \ldots, V_{L-1,i}$ for each training input $x_i$, where each $V_{j,i} \in \mathbb{R}^r$ serves as an approximation of the output of the $j$-th hidden layer for the $i$-th input. This leads us to reformulate the objective as:
$$\min_{\mathbf{W}, \mathbf{V}} F(\mathbf{W}, \mathbf{V}) \coloneqq \sum_{i=1}^{n} \left[ \|W_L V_{L-1,i} - y_i\|^2 + \gamma \sum_{j=1}^{L-1} \|\sigma(W_j V_{j-1,i}) - V_{j,i}\|^2 \right], \tag{2}$$
where $\gamma > 0$ is a regularization hyperparameter, $V_{0,i} \coloneqq x_i$, $\mathbf{W} = (W_1, \ldots, W_L)$, and $\mathbf{V}$ denotes the collection of all auxiliary variables. In this formulation, the second term in (2) quantifies the layer-wise reconstruction loss, measuring how well each auxiliary variable $V_{j,i}$ approximates the true output of the $j$-th layer, and the first term corresponds to the prediction error at the output layer. By construction, if $(\mathbf{W}^*, \mathbf{V}^*)$ satisfies $F(\mathbf{W}^*, \mathbf{V}^*) = 0$, then the corresponding weight matrices $\mathbf{W}^*$ form a global minimizer of the original problem (1).

One of the key advantages of the reformulated objective in (2) is that it allows us to treat the optimization with respect to the weights of each layer $W_1, \ldots, W_L$ independently. This decoupling simplifies the optimization process and enables efficient implementation strategies, such as parallelization. Although various methods have been proposed for optimizing (2), we focus on a relatively simple yet effective scheme: we update the weight matrices $W_j$ and the auxiliary variables $V_{j,i}$ sequentially, starting from the output layer and proceeding backward through the network. Concretely, the update sequence is given by
$$W_L \to V_{L-1,i} \to W_{L-1} \to \cdots \to V_{1,i} \to W_1,$$
by using the objective funtion (2). We summarize the full algorithm considered in this paper in Algorithm 1. In the following, we provide a detailed explanation of each step of the algorithm.

---

[1]If $\sigma$ is not differentiable, we assume that $\sigma(x_1) - \sigma(x_2) \geq \alpha(x_1 - x_2)$ for any $x_1, x_2 \in \mathbb{R}$.

---
**Algorithm 1:** Block Coordinate Descent
---
**Input**            : $K$: outer iterations, $K_V$: inner iterations for $V_{j,i}$, $K_W$: inner iterations for $W_1$,

                 $\eta_V$: step size for $V_{j,i}$, $\eta_W^{(1)}$, $\eta_W^{(2)}$: step sizes for weight updates

**Initialization :** $(W_1)_{ab} \overset{\text{i.i.d.}}{\sim} \mathcal{N}(0, d_{\text{in}}^{-1})$,    $(W_j)_{ab} \overset{\text{i.i.d.}}{\sim} \mathcal{N}(0, r^{-1})$ for $j = 2, \ldots, L$.

                  Apply singular value bounding to $W_j$ for $j = 2, \ldots, L$ (see Algorithm 2).

                  Set $V_{0,i} \leftarrow x_i$, and $V_{j,i} \leftarrow \sigma(W_j V_{j-1,i})$ for $j = 1, \ldots, L-1$.

**1**   **for** $k \leftarrow 1$ **to** $K$ **do**

**2**      $W_L \leftarrow W_L - \eta_W^{(1)} \nabla_{W_L} \sum_{i=1}^{n} \|W V_{L-1,i} - y_i\|^2$;

**3**      **for** $i \leftarrow 1$ **to** $n$ **do**

**4**         $V_{L-1,i} \leftarrow V_{L-1,i} - \eta_V \nabla_{V_{L-1,i}} \|W V_{L-1,i} - y_i\|^2$;

**5**      **for** $j \leftarrow L-1$ **to** $2$ **do**

**6**         $W_j \leftarrow W_j - \gamma \eta_W^{(1)} \nabla_{W_j} \sum_{i=1}^{n} \|\sigma(W_j V_{j-1,i}) - V_{j,i}\|^2$;

**7**         **for** $i \leftarrow 1$ **to** $n$ **do**

**8**             **for** $k_{in} \leftarrow 1$ **to** $K_V$ **do**

**9**                $V_{j-1,i} \leftarrow V_{j-1,i} - \gamma \eta_V \nabla_{V_{j-1,i}} \|\sigma(W_j V_{j-1,i}) - V_{j,i}\|^2$;

**10**      **for** $k_{in} \leftarrow 1$ **to** $K_W$ **do**

**11**         $W_1 \leftarrow W_1 - \gamma \eta_W^{(2)} \nabla_{W_1} \sum_{i=1}^{n} \|\sigma(W_1 V_{0,i}) - V_{1,i}\|^2$;
---

**Remark 4.1.** The introduction of auxiliary variables $V_{j,i}$ slightly increases memory usage with the number of samples, but the computation for each block can be executed in parallel or distributed across devices. In practice, this allows the method to scale efficiently even for moderately large datasets.

**Initialization:** Each weight matrix $W_j$ is initialized with Gaussian entries: $\mathcal{N}(0, d_{\text{in}}^{-1})$ for $W_1$, and $\mathcal{N}(0, r^{-1})$ for $j \geq 2$. For layers $j = 2, \ldots, L$, we apply *singular value bounding* (SVB) [18] by computing the singular value decomposition $W_j = U\Sigma V$, clipping singular values in $\Sigma$ to the range $[s_1, s_2]$, and reconstructing $W_j = U\Sigma' V$. The detailed implementation is provided in Algorithm 2 and is deferred to Appendix A due to space limitations[2].

Originally introduced to stabilize gradient-based training, SVB also enhances BCD performance. It improves the conditioning of the hidden layer loss $\|\sigma(W_j V_{j-1,i}) - V_{j,i}\|^2$ and mitigates large activations by constraining the operator norm of $W_j$. Auxiliary variables $V_j$ are then initialized exactly as $V_{j,i} = \sigma(W_j V_{j-1,i})$   for all $j = 1, \ldots, L-1$, $i = 1, \ldots, n$, This approach ensures zero hidden loss at initialization and promotes faster convergence.

**Update of $V$:** We update the auxiliary variables $V_{j,i}$ via gradient descent with a common step size $\eta_V$, applying multiple iterations per update. For $V_{L-1,i}$, we minimize the output loss:

$$V_{L-1,i} \leftarrow V_{L-1,i} - \eta_V \nabla_{V_{L-1,i}} \|W_L V_{L-1,i} - y_i\|^2,$$

which corresponds to solving the linear system $W_L V_{L-1,i} = y_i$. A unique solution exists if $W_L \in \mathbb{R}^{d_{\text{out}} \times r}$ has full row rank.

For $j = 2, \ldots, L-1$, we update $V_{j-1,i}$ using the hidden layer loss:

$$V_{j-1,i} \leftarrow V_{j-1,i} - \gamma \eta_V \nabla_{V_{j-1,i}} \|\sigma(W_j V_{j-1,i}) - V_{j,i}\|^2.$$

Assuming $\sigma$ satisfies Assumption 3.2, its strict monotonicity implies invertibility. Hence, updating $V_j$ approximates solving $W_{j+1} V_{j,i} = \sigma^{-1}(V_{j+1,i})$. Provided $W_{j+1} \in \mathbb{R}^{r \times r}$ is invertible, gradient descent with a properly chosen $\eta_V$ will converge to a solution.

Note that, unlike the hidden representations $V_{j,i}$ for $j < L-1$, the update of the final hidden representation $V_{L-1,i}$ (for the output layer) is performed only once per outer iteration, unlike the hidden layers. This asymmetry is intentional and theoretically justified (see Appendix D), since the output-layer subproblem is linear and converges in a single step.

---
[2]Unlike [18], which applies SVB throughout training, we restrict it to initialization. With a suitably small update step, the singular values of $W_j$ remain bounded, preserving the benefits of SVB without repeated enforcement.

**Remark 4.2.** In Algorithm 1, each hidden representation $V_{j,i}$ is updated using only the local loss $\|\sigma(W_j V_{j-1,i}) - V_{j,i}\|^2$, instead of solving the full BCD subproblem that also involves the adjacent layers. This simplified update is sufficient for convergence (see Theorem 5.1, since the hidden-layer losses become very small after each outer iteration, and further updates would have little effect on the overall optimization.

**Update of $W$:** Each weight matrix $W_j$ is updated using its corresponding loss term. For the output layer, the loss is $\sum_{i=1}^{n} \|W_L V_{L-1,i} - y_i\|^2$, while for hidden layers $j = 1, \ldots, L-1$, we use $\sum_{i=1}^{n} \|\sigma(W_j V_{j-1,i}) - V_{j,i}\|^2$.

Weights $W_j$ for $j = 2, \ldots, L$ are updated once per iteration using a step size $\eta_W^{(1)}$ (line 2 and 6):

$$W_L \leftarrow W_L - \eta_W^{(1)} \nabla_{W_L} \sum_{i=1}^{n} \|W_L V_{L-1,i} - y_i\|^2,$$

$$W_j \leftarrow W_j - \gamma \eta_W^{(1)} \nabla_{W_j} \sum_{i=1}^{n} \|\sigma(W_j V_{j-1,i}) - V_{j,i}\|^2.$$

In contrast, $W_1$ is updated multiple times per round using a different step size $\eta_W^{(2)}$ (line 11):

$$W_1 \leftarrow W_1 - \gamma \eta_W^{(2)} \nabla_{W_1} \sum_{i=1}^{n} \|\sigma(W_1 V_{0,i}) - V_{1,i}\|^2.$$

This asymmetry is essential for convergence. Since both $W_j$ and $V_{j-1,i}$ are updated for $j \geq 2$, a single weight update suffices to ensure the hidden loss decreases linearly by applying multiple updates to the auxiliary variables $V_{j-1,i}$, assuming the singular values of $W_j$ remain bounded. However, multiple updates are unnecessary and may destabilize training when $n > r$, as exact minimizers may not exist. For $W_1$, fixed inputs $V_{0,i} = x_i$ enable linear convergence under Assumption 3.1 ($d_{\text{in}} \geq n$, $\text{rank}(X) = n$), ensuring a global minimizer exists and justifying repeated updates.

**Remark 4.3.** Unlike prior BCD-based approaches that incorporate regularization or proximal updates [43, 19, 29], our method uses plain gradient descent. Though the convergence analysis is tailored to this setting, the framework extends naturally to alternative loss functions, classification tasks, and regularized objectives, as discussed in Appendix B. In addition, the same framework can be extended to handle non-bijective activations such as ReLU by incorporating skip connections and non-negative projections (see Section 6).

## 5 Global Convergence of Block Coordinate Descent

In this section, we demonstrate that block coordinate descent (BCD) applied to neural networks with activation functions satisfying Assumption 3.2 converges to global minima. That is, the objective value $F$ can be made arbitrarily small through the proposed training procedure. We focus first on the single-output case where $d_{\text{out}} = 1$. The extension to the multi-output case is discussed in Appendix C. Additionally, for the single-output setting, we derive a generalization error bound under the assumption of i.i.d. data, utilizing the Rademacher complexity framework.

### 5.1 Global Convergence with Monotonically Increasing Activation

We consider the case of single-output regression ($d_{\text{out}} = 1$), for which the objective function takes the form:

$$\min_{\mathbf{W},\mathbf{V}} F(\mathbf{W},\mathbf{V}) := \sum_{i=1}^{n} \left[ (W_L V_{L-1,i} - y_i)^2 + \gamma \sum_{j=1}^{L-1} \|\sigma(W_j V_{j-1,i}) - V_{j,i}\|^2 \right]. \tag{3}$$

We now state the first main result, the global convergence of BCD with activation satisfying Assumption 3.2.

**Theorem 5.1.** *Suppose that activation $\sigma$ satisfies Assumption 3.2. Let $s := \sigma_{\min}(X)$ denote the smallest singular value of the data matrix $X$. Let $R_i = |W_L V_{L-1,i} - y_i|$ be the initial residual at the output layer, and define $R := \sum_{i=1}^n R_i^2$, $R_{\max} := \max_i R_i$, and $C_K := \left(\frac{2}{\alpha}\right)^L \left(4R_{\max}\eta_V + \frac{2}{2-\alpha}\sqrt{\epsilon}\right)$. Then, there exists a constant $C_V > 0$ such that under $(s_1, s_2) = (\frac{3}{4}, \frac{5}{4})$, $\eta_V \leq \frac{\alpha}{16\gamma\ell^4}$, $\eta_W^{(1)} \leq \frac{\eta_V^{-1}}{8\sqrt{r}C_V K}\left(\frac{\alpha}{2}\right)^L$, $\eta_W^{(2)} \leq \frac{1}{\gamma\ell^4 \cdot \max_i \|x_i\|^2}$, and $K = \left\lceil \frac{2}{\eta_V} \log\left(\frac{3R}{\epsilon}\right) \right\rceil$, $K_V = \left\lceil \frac{1}{\gamma\alpha\ell\eta_V} \log\left(\frac{48\gamma\ell^2(L-2)rnC_K^2}{\alpha^2\epsilon}\right) \right\rceil$, $K_W = \left\lceil \frac{1}{4\gamma s\alpha^2\eta_W^{(2)}} \log\left(\frac{3\ell^2 \cdot \max_i \|x_i\|^2 rnC_K^2}{\alpha^2 s^2\epsilon}\right) \right\rceil$, it holds*
$$F(\mathbf{W}, \mathbf{V}) \leq \epsilon,$$
*where $\mathbf{W} = (W_1, \ldots, W_L)$ and $\mathbf{V} = (V_{1,1}, \ldots, V_{L-1,n})$ are the output of Algorithm 1.*

The proof is provided in Appendix D. Theorem 5.1 establishes that the proposed BCD algorithm provably converges to a global minimum. In particular, for any arbitrary accuracy level $\epsilon > 0$, the algorithm guarantees that the objective function value can be made less than $\epsilon$. While the definitions of the inner and outer loop iteration counts $K$, $K_V$, and $K_W$ are somewhat technical, the total number of gradient computations required to reach an $\epsilon$-accurate solution is bounded by $\tilde{O}\left(K(LK_V + K_W)\right) = \tilde{O}\left(nL \log^2\left(\frac{1}{\epsilon}\right)\right)$.

The proof is divided into two key parts: (i) the output-layer loss linearly decreases monotonically across outer iterations, and (ii) the hidden-layer losses remain sufficiently small at the end of each iteration. Further details are deferred to Appendix D.

It is important to emphasize that the convergence guarantees provided in Theorem 5.1 fall outside the scope of the neural tangent kernel (NTK) regime [17], among other related frameworks. Specifically, while the NTK regime assumes that the network parameters remain nearly constant throughout training, our analysis accommodates settings in which the parameters may evolve by a constant magnitude, i.e., change order $\Omega(1)$.

**Remark 5.2.** While Assumption 3.1 requires the data matrix $X \in \mathbb{R}^{n \times d_{in}}$ to have full row rank, we note that the proposed algorithm remains well-behaved even when $n > d_{in}$. In this case, the residual error at the first layer does not vanish completely but remains bounded, leading to an effective error level $\epsilon_{\text{total}} = \epsilon + \delta_1$, where $\delta_1$ represents the first-layer approximation error. When $X$ approximately spans the relevant subspace for $V_{1,i}$, $\delta_1$ is small, and the convergence behavior is qualitatively similar to the $n \leq d_{\text{in}}$ regime.

## 5.2 Generalization Error Bound

The objective of this subsection is to demonstrate that the BCD algorithm described in Algorithm 1 not only enjoys strong convergence guarantees but also achieves favorable generalization performance. To this end, we make the following assumption on the data distribution:

**Assumption 5.3.** The training sample $\{(x_i, y_i)\}_{i=1}^n$ is independently sampled from a distribution $(x, y) \sim P$. Under the distribution $P$, it holds that $\|x\| \leq B_X$ and $|y| \leq B_Y$ almost surely.

This assumption is commonly used in generalization error analysis. The first part specifies the i.i.d. nature of the data, while the boundedness conditions ensure that the loss remains controlled with high probability. We next define the generalization gap:

**Definition 5.4** (Generalization gap). The generalization gap is defined as the difference between training and test error, that is,

$$\mathsf{Gap} := \mathbb{E}_{(x,y)\sim P}\left[\left(\hat{f}_{\mathrm{NN}}(x) - y\right)^2\right] - \frac{1}{n}\sum_{i=1}^n \left(\hat{f}_{\mathrm{NN}}(x_i) - y_i\right)^2$$

We now present the main result of this subsection:

**Theorem 5.5.** *Let $\hat{f}_{NN}$ be the output of Algorithm 1 under the same condition as Theorem 5.1. Then, under Assumption 5.3, with probability at least $1 - \delta$ over the training samples $\{(x_i, y_i)\}_{i=1}^n$, the generalization gap satisfies*

$$\mathsf{Gap} \leq \tilde{O}\left(\frac{\|X\|}{n}(B_Y + 2^L\ell^{L-1}B_X)d_{in}^{\frac{1}{2}}L^{\frac{3}{2}}(2r)^{\frac{L}{2}}\log r + (B_Y + 2^L\ell^{L-1}B_X)^2\sqrt{\frac{\log(1/\delta)}{n}}\right).$$

The proof of Theorem 5.5 is provided in Appendix E. This result establishes that deep neural networks trained via block coordinate descent not only converge to global minima but also generalize well, extending guarantees beyond the NTK regime.

The derivation is based on the generalization analysis framework introduced by [8], which bounds the generalization gap in terms of the spectral norms of the weight matrices. As discussed in the previous sections, our convergence analysis guarantees that the spectral norm of each $W_j$ remains bounded throughout training. Combining this boundedness with the Rademacher complexity-based bounds in [8], we obtain a high-probability upper bound on the generalization error of trained networks.

# 6 ReLU Activation

In this section, we propose a modified BCD algorithm tailored explicitly for neural networks with the ReLU activation function, defined as $\sigma(x) := \max\{x, 0\}$. This setting is excluded from the analysis in Theorem 5.1 due to the violation of Assumption 3.2, which requires strict monotonicity. The primary difficulty in handling ReLU lies in its non-negative range. To achieve zero hidden layer loss of the form $\|\sigma(W_j V_{j-1}) - V_j\|^2$, we must ensure that $V_j$ does not contain negative entries—otherwise, the approximation cannot reach zero due to the non-negativity constraint imposed by ReLU. This necessitates a modification to the original BCD algorithm (Algorithm 1).

## 6.1 BCD for Neural Networks with Skip Connection

To address this issue, we employ a residual network (ResNet) architecture [16], incorporating skip connections into the model. This modifies the BCD objective as follows:

$$\min_{\mathbf{W}, \mathbf{V}} F(\mathbf{W}, \mathbf{V}) := \sum_{i=1}^{n} \Big[ (W_L V_{L-1,i} - y_i)^2 + \gamma \sum_{j=2}^{L-1} \|\sigma(W_j V_{j-1,i}) + V_{j-1,i} - V_{j,i}\|^2$$
$$+ \gamma \|\sigma(W_1 V_{0,i}) - V_{1,i}\|^2 \Big],$$

where the hidden layer loss now includes the skip connection term $V_{j-1,i}$, modifying the structure compared to the original formulation in (3). To guarantee that the auxiliary variables $V_{j,i}$ remain within the feasible range of ReLU outputs, we introduce non-negative projection steps of the form $V_{j,i}^+ := \max\{V_{j,i}, 0\}$. The complete algorithm, incorporating skip connections and projections, is detailed in Algorithm 3, which is deferred to Appendix A due to space limitations. In the following, we provide a detailed explanation of this modified procedure.

The initialization and update procedures for weight matrices remain unchanged between Algorithm 1 and Algorithm 3. However, several modifications are introduced to accommodate the ReLU activation. First, Algorithm 3 includes a non-negative projection $V \mapsto V^+$ applied to each $V_{j,i}$ after the inner loop, ensuring the equation $\|\sigma(W_j V_{j-1,i}) - V_{j,i}\|^2 = 0$ is solvable by aligning with the non-negative range of ReLU. Second, in contrast to Algorithm 1, the output layer weights $W_L$ are held fixed during training. This design ensures solvability of the equation $W_L V_{L-1,i} = y_i$ under the constraint $V_{L-1,i} \geq \mathbf{0}$. The following lemma formalizes this condition:

**Lemma 6.1.** *Suppose the vector $W_L$ contains both positive and negative entries. Then, for any $y_i$, there exists a non-negative vector $V_{L-1,i}$ such that $W_L V_{L-1,i} = y_i$.*

Lemma 6.1 ensures solvability if $W_L$ has mixed-sign entries, a condition increasingly probable as hidden layer width $r$ grows. Specifically, the probability is $1 - \frac{1}{2^{r-1}}$ under Gaussian initialization. Moreover, we provide a concentration bound on the positive and negative components of $W_L$, relevant to convergence rates:

**Lemma 6.2.** *Let $W_L^\top \sim \mathcal{N}(0, r^{-1} I_r)$, $w_+ := \max\{W_L, \mathbf{0}^\top\}$, and $w_- := \min\{W_L, \mathbf{0}^\top\}$. Then, for any $\delta > 0$, with probability at least $1 - 2\delta$, $w_{\min}^2 := \|w_+\|^2 \wedge \|w_-\|^2 \geq \frac{1}{2} - \sqrt{\frac{8 \log(2/\delta)}{r}}$ holds.*

To simplify analysis, we fix $W_L$ throughout training, since extending the bound in Lemma 6.2 to dynamic updates across iterations is non-trivial.

The update of $W_1$ is performed via gradient descent (line 12). Notably, the ReLU activation is omitted in this update. Since the projection step ensures $V_{1,i} \geq \mathbf{0}$, solving $\sigma(W_1 V_{0,i}) = V_{1,i}$ reduces to

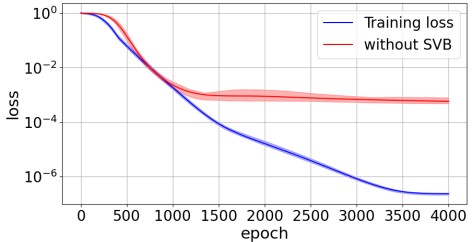
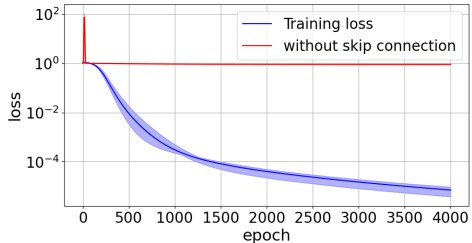

Figure 2: Training loss of Algorithm 1 with LeakyReLU activation ($\alpha = 0.5$).

Figure 3: Training loss of Algorithm 3 with ReLU activation.

solving the linear system $W_1 V_{0,i} = V_{1,i}$, which is more tractable. As in the previous section, we consider the single-output case $d_{\text{out}} = 1$. We now present the convergence result for Algorithm 3, applied to ReLU networks with skip connections.

**Theorem 6.3.** *Define $s$, $R_i$, $R_{\max}$, and $C_K$ as in Theorem 5.1. Then, there exists a constant $C_V > 0$ such that under $(s_1, s_2) = (0, \frac{1}{4})$, $\eta_V \leq \frac{1}{2w_{\min}^2} \wedge \frac{1}{12\gamma}$, $\eta_W^{(1)} \leq \frac{\eta_V^{-1}}{24\sqrt{r}C_V K} \left(\frac{2}{3}\right)^L$, $\eta_W^{(2)} \leq \frac{1}{2 \cdot \max_i \|x_i\|^2}$, and $K = \left\lceil \frac{1}{4\eta_V w_{\min}^2} \log\left(\frac{3R}{\epsilon}\right) \right\rceil$, $K_V = \left\lceil \frac{3}{4\gamma\eta_V} \log\left(\frac{245(L-2)rnC_K^2}{3\epsilon}\right) \right\rceil$, $K_W = \left\lceil \frac{1}{4\gamma s^2 \eta_W^{(2)}} \log\left(\frac{3\max_i \|x_i\|^2 C_K^2}{s^2\epsilon}\right) \right\rceil$, it holds*

$$F(\mathbf{W}, \mathbf{V}) \leq \epsilon,$$

*where $\mathbf{W} = (W_1, \ldots, W_L)$ and $\mathbf{V} = (V_{1,1}, \ldots, V_{L-1,n})$ are the output of Algorithm 3.*

The proof is provided in Appendix F. This result establishes a global convergence guarantee for BCD applied to deep neural networks with ReLU activation, under the skip-connection architecture.

## 7 Numerical Experiment

In this section, we conduct numerical experiments to empirically validate the theoretical results established in Sections 4 and 6. Specifically, we confirm that the BCD algorithms for networks with (i) strictly monotonically increasing activation functions (Algorithm 1) and (ii) ReLU activation with skip connections (Algorithm 3) successfully converge to global minima on a synthetic dataset. All experiments were conducted using Google Colab with a T4 GPU. Each experiment is independently repeated five times. We report the mean training loss along with standard deviation bands.

### 7.1 Monotonically Increasing Activation

We first evaluate BCD using a strictly monotonically increasing activation function. We apply Algorithm 1 to a neural network with four hidden layers of width $r = 30$, using LeakyReLU activation defined by $\sigma(x) = \max\{x, 0.5x\}$, which satisfies Assumption 3.2 with $\alpha = 0.5$ and $\ell = 1$. The training data consists of $n = 500$ samples generated from a teacher network with one hidden layer and the same activation. Each input $x_i \in \mathbb{R}^{600}$ is sampled from a standard Gaussian distribution, and the output $y_i$ is computed by the teacher network. We set the hyperparameters to $K_V = K_W = 100$, and all step sizes $\eta_V = \eta_W^{(1)} = \eta_W^{(2)} = 1$.

The results are shown in Figure 2. The blue curve represents the training loss. The training loss monotonically decreases, and the layer-wise residuals remain small, which aligns with our theoretical findings. To assess the effect of singular value bounding (SVB, Algorithm 2), we conduct an ablation experiment where the network is trained without applying SVB. The red curve in Figure 2 shows that, in this case, the loss stagnates around $10^{-3}$. This demonstrates that SVB contributes not only to theoretical convergence but also to practical training stability and effectiveness.

To further examine the scalability of the proposed BCD algorithm, we conducted additional experiments on deeper networks ($L = 8, 12$); the results consistently showed monotonic loss decrease across layers. Detailed results and plots are provided in Appendix G.

## 7.2 ReLU Activation

We next evaluate Algorithm 3 on a ReLU-activated network with skip connections. We use the same architecture as in the previous subsection: four hidden layers of width $r = 30$, with $n = 500$ training samples in $\mathbb{R}^{600}$, generated from a teacher network with ReLU activation. As before, we set $K_V = K_W = 100$ and step sizes $\eta_V = \eta_W^{(1)} = \eta_W^{(2)} = 1$.

Figure 3 shows the results. The blue curve represents the training loss using skip connections. As expected, the loss decreases monotonically, and the internal residuals remain small. To demonstrate the importance of skip connections, we also train a network without them. The red curve in Figure 3 shows that training fails to converge in this case due to the ReLU non-negativity constraint, which makes it challenging to match intermediate representations. This supports our theoretical conclusion that skip connections are crucial for achieving global convergence under ReLU activation.

## 8 Conclusion

In this work, we proposed a block coordinate descent (BCD) framework for training deep neural networks and established global convergence guarantees under strictly monotonically increasing activation functions. Our analysis demonstrated that the proposed method achieves arbitrarily small training loss, and we further derived a generalization bound based on Rademacher complexity. To address the challenges posed by non-monotonic activations such as ReLU, we introduced a modified BCD algorithm that incorporates skip connections and non-negative projections. This variant ensures global convergence even in the presence of ReLU activations, thereby extending the applicability of BCD to widely used modern architectures. Extensive numerical experiments corroborated our theoretical findings, showing that the proposed algorithms perform effectively in practice for both monotonic and ReLU activation functions.

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

## A   Omitted Pseudocode

In this section, we provide pseudocode for two procedures that were omitted in the main paper due to space limitations. Specifically, we describe the Singular Value Bounding (SVB, Algorithm 2) and the Block Coordinate Descent algorithm for ReLU activation (Algorithm 3).

---

**Algorithm 2:** Singular Value Bounding (SVB)

---

**Input**   : $W_j$: weight matrix, $(s_1, s_2)$: lower and upper bounds on singular values
**Output** : Regularized matrix with bounded singular values
1 Compute the singular value decomposition: $(U, \Sigma, V) \leftarrow \mathrm{SVD}(W_j)$;
2 **foreach** *singular value $s$ in the diagonal of $\Sigma$* **do**
3   $\quad$ $s \leftarrow \max\{s_1, \min\{s_2, s\}\}$ ;                              // Clip $s$ to the interval $[s_1, s_2]$
4 **return** $U\Sigma V^\top$

---

**Algorithm 3:** Block Coordinate Descent for ReLU Activation

---

**Input**           : $K$: outer iterations, $K_V$: inner iterations for $V_{j,i}$, $K_W$: inner iterations for $W_1$,
$\qquad\qquad$ $\eta_V$: step size for $V_{j,i}$, $\eta_W^{(1)}, \eta_W^{(2)}$: step sizes for weight updates
**Initialization** : $(W_1)_{ab} \overset{\text{i.i.d.}}{\sim} \mathcal{N}(0, d_{\text{in}}^{-1})$,   $(W_j)_{ab} \overset{\text{i.i.d.}}{\sim} \mathcal{N}(0, r^{-1})$ for $j = 2, \ldots, L$.
$\qquad\qquad$ Apply singular value bounding to $W_j$ for $j = 2, \ldots, L$ (see Algorithm 2).
$\qquad\qquad$ Set $V_{0,i} \leftarrow x_i$,
$\qquad\qquad$ Set $V_{1,i} \leftarrow \sigma(W_1 V_{0,i})$.
$\qquad\qquad$ Set $V_{j,i} \leftarrow \sigma(W_j V_{j-1,i}) + V_{j-1,i}$ for $j = 2, \ldots, L-1$.
1 **for** $k \leftarrow 1$ **to** $K$ **do**
2   $\quad$ **for** $i \leftarrow 1$ **to** $n$ **do**
3   $\quad\quad$ $V_{L-1,i} \leftarrow V_{L-1,i} - \eta_V \nabla_{V_{L-1,i}} \|W_L V_{L-1,i} - y_i\|^2$;
4   $\quad\quad$ $V_{L-1,i} \leftarrow (V_{L-1,i})^+$;
5   $\quad$ **for** $j \leftarrow L-1$ **to** $2$ **do**
6   $\quad\quad$ $W_j \leftarrow W_j - \gamma \eta_W^{(1)} \nabla_{W_j} \sum_{i=1}^n \|\sigma(W_j V_{j-1,i}) + V_{j-1,i} - V_{j,i}\|^2$;
7   $\quad\quad$ **for** $i \leftarrow 1$ **to** $n$ **do**
8   $\quad\quad\quad$ **for** $k_{in} \leftarrow 1$ **to** $K_V$ **do**
9   $\quad\quad\quad\quad$ $V_{j-1,i} \leftarrow V_{j-1,i} - \gamma \eta_V \nabla_{V_{j-1,i}} \|\sigma(W_j V_{j-1,i}) + V_{j-1,i} - V_{j,i}\|^2$;
10  $\quad\quad\quad$ $V_{j-1,i} \leftarrow (V_{j-1,i})^+$;
11  $\quad$ **for** $k_{in} \leftarrow 1$ **to** $K_W$ **do**
12  $\quad\quad$ $W_1 \leftarrow W_1 - \gamma \eta_W^{(2)} \nabla_{W_1} \sum_{i=1}^n \|W_1 V_{0,i} - V_{1,i}\|^2$;

---

## B   Discussion of Extension

As mentioned in Remark 4.3, our convergence analysis is based on a simple variant of block coordinate descent (BCD), where gradient descent is applied to minimize the standard squared loss. In this section, we discuss possible extensions of the proposed algorithms (Algorithms 1 and 3) to broader settings.

**General Loss Functions.**   A natural and practically relevant extension is to replace the squared loss with a general loss function $\ell(\cdot, \cdot)$. This modification allows the framework to go beyond regression and encompass classification and other supervised learning tasks. Under this extension, the objective becomes:

$$\min_{\mathbf{W}, \mathbf{V}} F(\mathbf{W}, \mathbf{V}) := \sum_{i=1}^n \left[ \ell(W_L V_{L-1,i}, y_i) + \gamma \sum_{j=1}^{L-1} \|\sigma(W_j V_{j-1,i}) - V_{j,i}\|^2 \right],$$

where only the output-layer loss differs from the original formulation in (3).

The hidden layer loss term remains unchanged, so the analysis in Theorem 5.1 still applies to those layers. Therefore, to establish convergence in this generalized setting, it suffices to prove that the output layer subproblem involving $W_L$ and $V_{L-1}$ converges globally under the new loss $\ell$.

For example, if $\ell(\cdot, \cdot)$ is strongly convex (e.g., logistic or cross-entropy loss), standard results in convex optimization can be used to ensure convergence. A typical case is the cross-entropy loss for multi-class classification:

$$\ell(W_L V_{L-1,i}, y_i) = -\sum_{c=1}^{d_{\text{out}}} y_{ic} \log \left( \frac{\exp((W_L V_{L-1,i})_c)}{\sum_{c'=1}^{d_{\text{out}}} \exp((W_L V_{L-1,i})_{c'})} \right),$$

which is widely used for $d_{\text{out}}$-class classification tasks.

Thus, although our main analysis focuses on regression using the squared loss, the BCD framework and associated convergence guarantees can be extended to classification problems and other supervised learning settings by appropriately modifying the output-layer loss.

**Different Activation Functions Across Layers.** While our analysis assumes that all layers share the same activation function $\sigma$, it is straightforward to extend the results to the case where each layer uses a distinct activation $\sigma_j$, provided that each $\sigma_j$ satisfies Assumption 3.2. In this case, the convergence proof in Theorem 5.1 remains valid by replacing $\sigma$ with $\sigma_j$ in the convergence argument corresponding to the $j$-th layer.

**Alternative Initialization Schemes.** In Algorithm 1, we initialize the weights $W_j$ using Gaussian distributions and apply singular value bounding (SVB), while the auxiliary variables $V_{j,i}$ are initialized exactly as $V_{j,i} = \sigma(W_{j-1} V_{j-1,i})$. However, global convergence only requires that the regularity condition from Lemma D.4 be maintained throughout training. Hence, the initialization scheme is flexible. In particular, Gaussian initialization is not necessary—alternative methods such as Xavier initialization, which uses a uniform distribution with appropriately scaled bounds, can also be applied as long as they yield well-conditioned weight matrices.

**Activations Violating Assumption 3.2.** We now consider the use of activation functions that do not satisfy Assumption 3.2, including those beyond ReLU. Our analysis heavily relies on the monotonicity of the activation function to ensure that the optimization landscape avoids undesirable local minima caused by vanishing gradients (e.g., points where $\sigma' = 0$). Without monotonicity, there is no guarantee that gradient-based updates will escape such critical points, which may result in failure to reach the global minimum.

That said, some commonly used monotonic activations that violate Assumption 3.2, such as sigmoid and tanh, are still of interest. The key challenge in analyzing these functions lies in their bounded output ranges:

- Sigmoid: $\sigma(x) = \frac{1}{1+\exp(-x)} \in (0, 1)$,
- Tanh: $\sigma(x) = \frac{\exp(x)-\exp(-x)}{\exp(x)+\exp(-x)} \in (-1, 1)$.

In these cases, it becomes essential to ensure that the auxiliary variables $V_{j,i}$ remain within the output range of the corresponding activation function. This is analogous to the ReLU case, where we enforce non-negativity through projection. As discussed in Section 6, we addressed this challenge for ReLU using skip connections. Similar techniques—such as range-aware projections or bounded initialization—may be required to extend BCD to these bounded activations.

**Training Loss with Regularization Terms.** A line of work on BCD methods considers regularized training objectives, where the loss function includes additional penalty terms. In this setting, the objective takes the form:

$$\min_{\mathbf{W}, \mathbf{V}} F(\mathbf{W}, \mathbf{V}) := \sum_{i=1}^{n} \Bigg[ (W_L V_{L-1,i} - y_i)^2 + r_W(W_L)$$

$$+ \gamma \sum_{j=1}^{L-1} \left( \|\sigma(W_j V_{j-1,i}) - V_{j,i}\|^2 + r_W(W_j) + r_V(V_j) \right) \Bigg],$$

where $r_W$ and $r_V$ denote regularization terms for weights and hidden representations, respectively.

In this setting, additional gradient terms appear in the update steps for $W_j$ and $V_{j,i}$, corresponding to the gradients of $r_W$ and $r_V$. While the inclusion of regularization complicates the convergence analysis, global convergence can still be established under certain conditions—specifically, when the regularizers are strongly convex. A common example is Tikhonov (or $\ell_2$) regularization.

However, incorporating regularization introduces a trade-off between optimization and generalization. In particular, to derive a generalization error bound analogous to Theorem 5.5, it is necessary to carefully analyze the gap in training loss introduced by the regularization terms. This involves quantifying how regularization affects the empirical risk and bounding its impact on the generalization gap.

## C    Extension to Multi-Dimensional Output

We now consider the case of multi-dimensional outputs, where the loss function is given by:

$$\min_{\mathbf{W},\mathbf{V}} F(\mathbf{W},\mathbf{V}) := \sum_{i=1}^{n} \left[ \|W_L V_{L-1,i} - y_i\|^2 + \gamma \sum_{j=1}^{L-1} \|\sigma(W_j V_{j-1,i}) - V_{j,i}\|^2 \right],$$

with $y_i \in \mathbb{R}^{d_{\mathrm{out}}}$ and $d_{\mathrm{out}} > 1$.

Compared to the scalar-output setting, the main challenge lies in analyzing the convergence of the output-layer parameters $W_L$ and $V_{L-1,i}$. When $\mathrm{rank}(W_L) \geq d_{\mathrm{out}}$, the linear system

$$W_L V_{L-1,i} = y_i \tag{4}$$

has solutions, and the convergence analysis follows similarly to Theorem 5.1 using standard gradient descent arguments.

However, when $d_{\mathrm{out}} > \mathrm{rank}(W_L)$, the system (4) may not admit a solution, making global convergence unattainable without additional assumptions. To address this, we introduce the following low-rank structure assumption on the labels:

**Assumption C.1** (Low-Rank Label Representation). There exists an integer $r < d_{\mathrm{out}}$ and a matrix $U_1 \in \mathbb{R}^{d_{\mathrm{out}} \times r}$ such that for all $i \in \{1, \ldots, n\}$, the label satisfies $y_i = U_1 z_i$ for some $z_i \in \mathbb{R}^r$.

Under Assumption C.1, the system (4) has a solution—for example, choosing $W_L = U_1$ and $V_{L-1,i} = z_i$. However, the question remains whether gradient descent can find such a solution in practice. To explore this, consider the gradient descent update for $W_L$, as in line 2 of Algorithm 1. For general $d_{\mathrm{out}}$, the update can be written as:

$$W_L^{(k)} = W_L^{(k-1)} \left( I - \eta_W \sum_{i=1}^{n} V_{L-1,i} V_{L-1,i}^{\top} \right) + \eta_W U_1 \sum_{i=1}^{n} z_i V_{L-1,i}^{\top}.$$

Here, the first term implies that with a sufficiently small step size $\eta_W$, the norm of $W_L$ decays exponentially in directions orthogonal to the span of $\{V_{L-1,i}\}$. Meanwhile, the second term injects components aligned with $U_1$. In particular, when the matrix $\sum_{i=1}^{n} z_i V_{L-1,i}^{\top} \in \mathbb{R}^{r \times r}$ is full rank, the expression

$$W_L = \eta_W U_1 \sum_{i=1}^{n} z_i V_{L-1,i}^{\top}$$

may serve as a solution to (4) if an appropriate inverse exists.

While this offers intuitive insight, rigorous convergence guarantees in the multi-output case are non-trivial. Fortunately, a recent result by [39] addresses this issue:

**Theorem C.2** (Theorem 1.1 in [39]). *Suppose $Y = [y_1, \ldots, y_n] \in \mathbb{R}^{d_{\mathrm{out}} \times n}$ satisfies Assumption C.1. Let $s_1$ and $s_r$ denote the smallest and largest singular values of $Y$, respectively. Assume all entries of $W_L$ and $V_{L-1,i}$ are initialized independently from $\mathcal{N}(0, \delta^2)$, where*

$$\delta = \tilde{O}\left( \frac{s_r}{\sqrt{r^3 s_1}(n + d_{\mathrm{out}})} \right).$$

*Then, using step size $\eta = O\left(\frac{s_r \delta^2}{r s_1}\right)$, gradient descent satisfies $\sum_{i=1}^n \|W_L V_{L-1,i} - y_i\|^2 \le \epsilon$ after*

$$O\left(\frac{1}{\eta s_1} \log\left(\frac{r s_r}{\epsilon}\right) + \frac{1}{\eta s_r} \log\left(\frac{s_r}{\epsilon}\right)\right)$$

*iterations.*

To apply Theorem C.2 to our BCD setting, two modifications are required relative to the proof of Theorem 5.1:

1. Replace the convergence argument for $W_L$ and $V_{L-1,i}$ with Theorem C.2, yielding the iteration bound from the theorem.

2. Adjust the initialization scheme: Theorem C.2 assumes Gaussian initialization for both $W_L$ and $V_{L-1,i}$, which differs from the exact-layer initialization $V_{j,i} = \sigma(W_{j-1} V_{j-1,i})$ used in Theorem 5.1.

As discussed in Appendix B, the exact-layer initialization mainly serves to ensure a small initial objective value, and our analysis can be extended to other initialization schemes. Therefore, by adopting Gaussian initialization and integrating Theorem C.2, the convergence guarantees of BCD can be extended to multi-output settings.

# D   Proof of Theorem 5.1

In this section, we provide the proof to Theorem 5.1. The key notion is the block-wise analysis. First, we provide the preliminary lemmas for the proof. After that, we prepare the block-wise analysis and combine them.

Throughout this section, we suppose that the conditions in Theorem 5.1 are satisfied.

## D.1   Preliminary Results

The following lemma immediately follows from the smoothness of the activation.

**Lemma D.1.** *Let $d \ge 1$ an integer. For any $x_1$, $x_2 \in \mathbb{R}^d$, it holds that $\|\sigma(x_1) - \sigma(x_2)\|^2 \le \ell^2 \|x_1 - x_2\|^2$.*

Next, by utilizing Assumption 3.2, we derive the following lemma.

**Lemma D.2.** *For activation function satisfying Assumption 3.2, for any $x$, $y \in \mathbb{R}$, there exists $\xi$ such that $\alpha \le \xi \le \ell$ and $\sigma(x + y) = \sigma(x) + \xi y$ hold.*

*Proof.* We first consider the case $y > 0$. Then, we have

$$\sigma(x + y) - \sigma(x) = \int_0^y \sigma'(x + t) dt \ge \alpha y.$$

The Lipschitz continuity of $\sigma$ gives $\sigma(x + y) \le \sigma(x) + \ell y$. Thus we get

$$\alpha \le \xi := \frac{\sigma(x + y) - \sigma(x)}{y} \le \ell,$$

which gives the conclusion.

The case $y < 0$ can be proven by substituting $x$ and $y$ in above discussion by $x + y$ and $-y$.

In the case $y = 0$ we can take arbitrary $\xi$ with $\alpha \le \xi \le \ell$ to satisfy the assertion. $\qquad\square$

This lemma gives the following proposition, which we utilize throughout the convergence analysis.

**Proposition D.3.** *For activation functions satisfying Assumption 3.2 and an integer $d > 1$, for any $x$, $y \in \mathbb{R}^d$, there exists a diagonal matrix $\Xi$ such that each diagonal entry $\Xi_{jj}$ of $A$ satisfies $\alpha < \Xi_{jj} < \ell$ and $\sigma(x + y) = \sigma(x) + \Xi y$.*

*Proof.* Note that by Lemma D.2, for each $j = 1, \dots, d$, there exists a $\Xi_{jj}$ satisfying $\sigma(x+y)_j = \sigma(x)_j + \Xi_{jj} y_j$. Then, $\Xi = \mathrm{diag}(\Xi_{11}, \dots, \Xi_{dd})$ satisfies the desired condition. $\square$

Next, we prove that the singular values of $W_j$ ($j = 2, \dots, L$) are upper and lower bounded during the training.

**Lemma D.4** (Regularity of weight matrix $W_j$ during training). *For $j = 2, \dots, L$, $\frac{1}{2} \le \lambda_{\min}^{1/2}(W_j W_j^T) \le \lambda_{\max}^{1/2}(W_j W_j^\top) \le 2$ always holds during the training.*

To obtain this lemma, we utilize the following fact:

**Lemma D.5** (Weyl's inequality for singular values). *Let $A \in \mathbb{R}^{d_1 \times d_2}$ be a real-valued matrix, then, for every matrix $\Delta \in \mathbb{R}^{d_1 \times d_2}$, it holds that*

$$\max_k |\sigma_k(A + \Delta) - \sigma_k(A)| \le \sigma_{\max}(\Delta),$$

*where $\sigma_k(A)$ denotes the $k$-th largest singular value of $A$ and $\sigma_{\max}(A)$ denotes its maximum singular value.*

*Proof of Lemma D.4.* By Lemma D.5, it suffices to show that every row $w$ of $W_j$ satisfies $\|\Delta w\| \le \frac{1}{4\sqrt{r}}$, where $\Delta w$ denotes the difference between $w$ at the start and end of the training. Indeed, this implies

$$\sigma_{\max}(\Delta W) = \lambda_{\max}^{\frac{1}{2}}(\Delta W \Delta W^\top) \le \sqrt{\sum_{p=1}^r \lambda_p(\Delta W \Delta W^\top)} \le \sqrt{\mathrm{Tr}(\Delta W \Delta W^\top)}$$

$$= \|\Delta W\|_F \le \frac{1}{4}.$$

Combining this bound with $\frac{3}{4} \le \sigma_{\min}(W_j) \le \sigma_{\max}(W_j) \le \frac{5}{4}$, which holds at the initialization, gives the conclusion.

To this end, we prove $\|\Delta w\| \le \frac{1}{4\sqrt{r}}$. This follows from

$$\eta_W^{(1)} \gamma \nabla_w \|\sigma(wV) - V'\|^2 = 2\eta_W^{(1)} \gamma \cdot \left\| \mathrm{diag}(\sigma'(wV)) V^\top (\sigma(wV) - V') \right\|$$

$$\le 2\eta_W^{(1)} \gamma \ell \lambda_{\max}^{1/2}(VV^\top) \cdot \|\sigma(wV) - V'\|$$

$$\le 2\eta_W^{(1)} \gamma \ell C_V \cdot \eta_V \left(\frac{2}{\alpha}\right)^L \le \frac{1}{4K\sqrt{r}},$$

where the second inequality follows from Lemma D.6 and the last inequality follows from the definition of $\eta_W^{(1)}$. $\square$

**Lemma D.6.** *Let $c_V := 2 \max_j \sum_{i=1}^n \|V_{j,i}\|^2$, where $V_j$s are the parameters at the initialization. Under the same settings as Theorem 5.1, let $C_V = c_V + \mathcal{O}((\gamma \eta_V \ell n K K_V)^2)$. Then, $\lambda_{\max}(V_j V_j^\top) \le C_V$ holds for $j = 1, \dots, L - 1$ during the training.*

*Proof.* First, we have

$$\lambda_{\max}(V_j V_j^\top) \le \sum_{j=1}^r \lambda_j(V_j V_j^\top) = \mathrm{tr}(V_j V_j^\top) = \mathrm{tr}\left(\sum_{i=1}^n V_{j,i} V_{j,i}^\top\right) = \sum_{i=1}^n \mathrm{tr}(V_{j,i} V_{j,i}^\top)$$

$$= \sum_{i=1}^n \|V_{j,i}\|^2. \qquad (5)$$

This implies that we only need to evaluate the norm of $V_{j,i}$s during the training. Remind that the update of $V_j$ is given by

$$V_{j,i} \leftarrow V_{j,i} - 2\gamma \eta_V W_j^\top D(\sigma(W_j V_{j,i}) - V_{j+1,i}),$$

where $D = \text{diag}(\sigma'(W_j V_{j,i}))$. Let $\Delta V_{j,i} := 2\gamma \eta_V W_j^\top D(\sigma(W_j V_{j,i}) - V_{j+1,i})$. Then, we have

$$\|\Delta V_{j,i}\| = 2\gamma \eta_V \|W_j^\top D(\sigma(W_j V_{j,i}) - V_{j+1,i})\|$$

$$\leq 2\gamma \eta_V \cdot \|W_j\|_{op} \|D\|_{op} \|\sigma(W V_{j,i}) - V_{j+1,i}\| \leq 4\gamma \ell \eta_V C_K,$$

where in the second inequality, we use $\|W_j\|_{op} = \lambda_{\max}^{1/2}(W_j W_j^\top) \leq 2$ from Lemma D.4, $\|D\|_{op} \leq \ell$, and $\|\sigma(W V_{j,i}) - V_{j+1,i}\| \leq C_K$ from (15) (Note that the objective function $\|\sigma(W V_{j,i}) - V_{j+1,i}\|^2$ is monotonically decreasing from Lemma D.8, (15) always holds). Since the total number of updating $V_{j,i}$ is $K \cdot K_V$, by using the triangle inequality we have

$$\|V_{j,i}\| \leq \|V_{j,i}^{init}\| + K \cdot K_V \cdot 4\gamma \ell \eta_V C_K,$$

where $\|V_{j,i}^{init}\|$ is the initial value of $V_{j,i}$.

Substituting this bound to (5), we obtain

$$\lambda_{\max}(V_j V_j^\top) \leq \sum_{i=1}^{n} \left( \|V_{j,i}^{init}\| + K \cdot K_V \cdot 4\gamma \ell \eta_V C_K \right)^2$$

$$\leq \sum_{i=1}^{n} 2 \left( \|V_{j,i}^{init}\|^2 + (K \cdot K_V \cdot 4\gamma \ell \eta_V C_K)^2 \right)$$

$$\leq c_V + \mathcal{O}((\gamma \eta_V \ell n K K_V)^2),$$

where we use the inequality $(a + b)^2 \leq 2a^2 + 2b^2$ in the second inequality. Thus, we obtain the conclusion. $\qquad\square$

## D.2 Analysis of Gradient Descent in a General Form

First, we introduce the key idea of analysis with general notations[3]. Let us consider the regression problem with an objective

$$F_{gen}(w) := \sum_{a=1}^{b} \left( \sigma(w^\top x_a) - y_a \right)^2, \tag{6}$$

where $w \in \mathbb{R}^d$ is a trainable parameter. Let $w' := w - \eta \nabla_w F_{gen}(w)$, where $w'$ denotes the parameter obtained by a single update of gradient descent with a step-size $\eta > 0$. Denote $X := (x_1, \dots x_b)^\top \in \mathbb{R}^{b \times d}$ and $Y := (y_1, \dots, y_b)^\top \in \mathbb{R}^b$. Then, $\sum_{a=1}^{b} \left( \sigma(w^\top x_a) - y_a \right)^2 = \|\sigma(Xw) - Y\|^2$ holds and a straightforward calculation shows $w' = w - 2\eta X^\top D(\sigma(Xw) - Y)$, where $D = \text{diag}((\sigma'(w^\top x_1), \dots, \sigma'(w^\top x_b)))$.

Now, we assume that there exists a unique optimal solution $w^*$ satisfying $F_{gen}(w^*) = 0$, i.e., $Y = \sigma(Xw^*)$. Then, we have

$$\|w' - w^*\|^2 = \|w - \eta \nabla_w F_{gen}(w) - w^*\|^2$$

$$= \|w - w^*\|^2 - 2\eta \nabla_w F_{gen}(w)^\top (w - w^*) + \eta^2 \|\nabla_w F_{gen}(w)\|^2$$

and

$$\nabla_w F_{gen}(w)^\top (w - w^*) = 2(\sigma(Xw) - \sigma(Xw^*))^\top DX(w - w^*)$$

$$= 2(\Xi X(w - w^*))^\top DX(w - w^*)$$

$$= 2(w - w^*)^\top X^\top \Xi DX(w - w^*)$$

$$\geq 2\lambda_{\min}(X^\top \Xi DX)\|w - w^*\|^2,$$

where $\Xi$ is a diagonal matrix determined by Proposition D.3 and $\lambda_{\min}(X^\top \Xi DX)$ is the smallest eigenvalue of $X^\top \Xi DX$.

Moreover, we have the upper bound of the gradient as

$$\|\nabla_w F_{gen}(w)\|^2 = \|2X^\top D(\sigma(Xw) - Y)\|^2 \leq 4\lambda_{\max}(DX^\top XD)\|\sigma(Xw) - Y\|^2,$$

where $\lambda_{\max}(DX^\top XD) \geq 0$ is the largest eigenvalue of the matrix $DX^\top XD$.

---

[3]Our analysis is similar to that in [40, 14].

### D.3 Block-wise Convergence Analysis

According to the observation in Appendix D.2, we provide the block-wise convergence analysis, that is, the convergence analysis of the $W_j$ and $V_j$ of the each layer.

**Update of $V_{j,i}$ ($j = 1, \ldots, L - 2$)** According to Algorithm 1, the update of $V_{j,i}$ ($j = 1, \ldots, L - 1$) is written by

$$V_{j,i} \leftarrow V_{j,i} - \gamma \eta_V \sum_{p=1}^{r} \nabla_{V_{j,i}} \left( \sigma(w_{j,p} V_{j,i}) - (V_{j+1,i})_p \right)^2, \tag{7}$$

where $w_{j,p}$ denotes the $p$-th row of the weight matrix of the $j$-th layer $W_j$ and $(V_{j+1,i})_p$ denotes the $p$-th component of $V_{j+1,i}$.

Despite the abuse of notation, we omit the layer index $j$ and the sample index $i$ for notational simplicity. We note that the analysis here can be independently applied to each layer and sample, as shown in the proof of the main theorem; hence, this abbreviation does not matter in the proof of Theorem 5.1. Then, (7) can be rewritten by

$$V \leftarrow V - \gamma \eta_V \sum_{p=1}^{r} \nabla_V \left( \sigma(w_p V) - V_p' \right)^2, \tag{8}$$

where we denote $V_p' := (V_{j+1,i})_p$.

Let $F_V(v) := \sum_{p=1}^{r} \left( \sigma(w_p v) - V_p' \right)^2 (= \|\sigma(Wv) - V'\|^2)$ and $V^{(0)}$ be the initial point of $V_j$ of the inner loop for each outer iteration (we also use abuse of notation here), and $V^{(k)}$ be the parameter obtained by $k$ iterations of the inner loop.

Under these settings, we first show the existence of global minima of $F_V$ as follows:

**Lemma D.7** (Existence of $v^*$). *Suppose that $\frac{1}{2} \leq \sigma_{\min}(W)$ and $\sigma_{max}(W) \leq 2$ hold. Let $\Delta v := \sigma(WV^{(0)}) - V'$. Then, there exists a unique $v$ satisfying $F_V(v^*) = 0$ and*

$$\left\| V^{(0)} - v^* \right\| \leq \frac{2}{\alpha} \|\Delta v\|.$$

*Proof.* Let $\Delta v := \sigma^{-1}(\sigma(WV^{(0)}) + \Delta v) - WV^{(0)}$. Then, it follows that $v^* = V^{(0)} + W^{-1} \Delta v$ since

$$\sigma \left( V^{(0)} + W^{-1} \Delta v \right) = \sigma \left( \sigma^{-1} \left( \sigma(WV^{(0)}) + \Delta v \right) \right) = \sigma \left( WV^{(0)} \right) + \Delta v.$$

Now, $\sigma^{(-1)}(\cdot)$ is $\frac{1}{\alpha}$-Lipschitz and satisfies $\sigma(0) = 0$. Then, we have $\|\Delta v\| \leq \frac{1}{\alpha} \|\Delta v\|$ and consequently

$$\left\| V^{(0)} - v^* \right\| = \left\| W^{-1} \Delta v \right\| \leq \left\| W^{-1} \right\|_{op} \cdot \|\Delta v\| \leq \frac{2}{\alpha} \|\Delta v\|.$$

This gives the assertion. $\qquad \square$

Next, by using the observation in Appendix D.2, we provide the convergence analysis to the update (8).

**Lemma D.8** (Convergence analysis of $V$). *Under the same condition as Theorem 5.1, it holds that*

$$\left\| \sigma(WV^{(k)}) - V' \right\|^2 \leq \frac{16\ell^2}{\alpha^2} \exp \left( -\frac{\alpha^2}{4} \gamma \eta_V k \right) \left\| \sigma \left( WV^{(0)} \right) - V' \right\|^2.$$

*Proof.* By letting $a \to p$, $b \to r$, $x_a \to w_p$, $y_a \to V_p'$ in (6), we have

$$\left\| V^{(k+1)} - v^* \right\|^2 \tag{9}$$

$$= \left\| V^{(k)} - v^* \right\|^2 - 2\gamma \eta_V \nabla_V F_V \left( V^{(k)} \right)^\top \left( V^{(k)} - V^* \right) + \gamma^2 \eta_V^2 \left\| \nabla_V F_V \left( V^{(k)} \right) \right\|^2.$$

The second term can be bounded by

$$\nabla_v F_V\left(V^{(k)}\right)^\top \left(V^{(k)} - v^*\right) \geq 2\lambda_{\min}\left(W^\top \Xi DW\right)\left\|V^{(k)} - v^*\right\|^2,$$

where $D = \operatorname{diag}\left((\sigma'(w_1 V^{(k)}), \ldots, \sigma'(w_r V^{(k)}))\right) \in \mathbb{R}^{r\times r}$. Then, we have

$$\lambda_{\min}\left(W^\top \Xi DW\right) \geq \alpha^2 \cdot \lambda_{\min}(W^\top W) \geq \frac{\alpha^2}{4},$$

where we use Lemma D.4 for the last inequality.

Moreover, the third term can be bounded by

$$\left\|\nabla_V F_V\left(V^{(k)}\right)\right\|^2 \leq 4\lambda_{\max}(DW^\top WD)\|\sigma(Xw) - \sigma(Xw^*)\|^2$$

$$\leq 4\ell^2 \lambda_{\max}(W^\top W)\left\|\sigma(WV^{(k)}) - \sigma(Wv^*)\right\|^2$$

$$\leq \ell^2 \cdot \ell^2 \|W\|_{op}^2 \left\|V^{(k)} - v^*\right\|^2 = 4\ell^4 \left\|V^{(k)} - v^*\right\|^2,$$

where we use Lemma D.4 in the third and last inequalities and Lemma D.1 in the third inequality.
By substituting these bounds to (9), we obtain

$$\left\|V^{(k+1)} - v^*\right\|^2$$

$$\leq \left\|V^{(k)} - v^*\right\|^2 - \frac{\alpha^2}{2}\gamma\eta_V \left\|V^{(k)} - v^*\right\|^2 + 4\gamma^2 \eta_V^2 \ell^4 \left\|V^{(k)} - v^*\right\|^2$$

$$\leq \left\|V^{(k)} - v^*\right\|^2 - \gamma\eta_V \left(\frac{\alpha^2}{2} - 4\gamma\eta_V \ell^4\right)\left\|V^{(k)} - v^*\right\|^2$$

$$\leq \left\|V^{(k)} - v^*\right\|^2 - \frac{\alpha^2}{4}\gamma\eta_V \left\|V^{(k)} - v^*\right\|^2 = \left(1 - \frac{\alpha^2}{4}\gamma\eta_V\right)\left\|V^{(k)} - v^*\right\|^2,$$

where we use $\eta_V \leq \frac{\alpha^2}{16\gamma\ell^4}$ in the last inequality. This implies

$$\left\|V^{(k)} - v^*\right\|^2 \leq \left(1 - \frac{\alpha^2}{4}\gamma\eta_V\right)^k \left\|V^{(0)} - v^*\right\|^2 \leq \exp\left(-\frac{\alpha^2}{4}\gamma\eta_V k\right)\left\|V^{(0)} - v^*\right\|^2,$$

where we use $1 - x \leq \exp(-x)$ in the first inequality, and hence,

$$\left\|\sigma(WV^{(k)}) - V'\right\|^2 \leq \ell^2 \left\|W\left(V^{(k)} - v^*\right)\right\|^2$$

$$\leq 4\ell^2 \left\|V^{(k)} - v^*\right\|^2$$

$$\leq 4\ell^2 \exp\left(-\frac{\alpha^2}{4}\gamma\eta_V k\right)\left\|V^{(0)} - v^*\right\|^2$$

$$\leq \frac{16\ell^2}{\alpha^2} \exp\left(-\frac{\alpha^2}{4}\gamma\eta_V k\right)\left\|\sigma\left(WV^{(0)}\right) - V'\right\|^2,$$

which gives the conclusion. □

Finally, we provide a lemma evaluating distance to global minima based on the objective value:

**Lemma D.9.** *Suppose that $F_V(v) \leq \epsilon$ holds. Then, $\|v - v^*\| \leq \frac{2}{\alpha}\sqrt{\epsilon}$.*

*Proof.* Since

$$\epsilon \geq F_V(v) = \|\sigma(Wv) - \sigma(Wv^*)\|^2 \geq \alpha^2 \|Wv - Wv^*\|^2 \geq \frac{1}{4}\alpha^2 \|v - v^*\|^2,$$

we obtain $\|v - v^*\| \leq \frac{2}{\alpha}\sqrt{\epsilon}$. □

**Update of $W_j$ ($j = 2, \ldots, L-1$)**  Let $w_{j,p} \in \mathbb{R}^r$ be the $p$-th row of the weight matrix $W_j$. Then, update of each $w_{j,p}$ is given by

$$w_{j,p} \leftarrow w_{j,p} - \gamma \eta_W^{(1)} \sum_{i=1}^{n} \nabla_{w_{j,p}} \left( \sigma(w_{j,p} V_{j,i}) - (V_{j+1,i})_p \right)^2, \tag{10}$$

For notational simplicity, we omit the layer index $j$ and the node index $p$. Namely, the update (10) is simply rewritten by

$$w \leftarrow w - \gamma \eta_W^{(1)} \sum_{i=1}^{n} \nabla_w (\sigma(w V_i) - V_i')^2,$$

where we denote $V_i := V_{j,i}$ and $V_i' := (V_{j+1,i})_p$. Let $F_W(w) := \sum_{i=1}^{n} (\sigma(w V_i) - V_i')^2 (= \|\sigma(wV) - V'\|^2)$ and $w^{(0)}$ be the initial point of $w_{j,p}$ of the inner loop for each outer iteration (we also use abuse of notation here), and $w^{(k)}$ be the parameter obtained by $k$ iterations of the inner loop. Against to the argument of $F_V$ in the above paragraph, $F_W$ have not a solution $w^*$ satisfying $F_W(w^*) = 0$ especially when $n > r$. However, we can still ensure that the objective value remains small during the update of $W_j$.

**Update of $W_1$**  Let $w_p \in \mathbb{R}^{d_{in}}$ the $p$-th row of the weight matrix $W_1$. Then, the update of each $w_p$ is given by

$$w_p \leftarrow w_p - \gamma \eta_W^{(2)} \sum_{i=1}^{n} \nabla_{w_l} \left( \sigma(w_p x_i) - (V_{1,i})_p \right)^2. \tag{11}$$

Despite the abuse of notation, we omit the node index $p$ for notational simplicity. Namely, the update (11) is simply rewritten by

$$w^{(k)} \leftarrow w^{(k-1)} - \gamma \eta_W^{(2)} \sum_{i=1}^{n} \nabla_w \left( \sigma\left( w^{(k-1)} x_i \right) - V_i \right)^2, \tag{12}$$

where we denote $V_i := (V_{1,i})_l$.

Let $F_W(w) := \sum_{i=1}^{n} (\sigma(w x_i) - V_i)^2 (= \|\sigma(wX) - V\|^2)$ and $w^{(0)}$ be the initial point of $w_j$ of the inner loop for each outer iteration (we also use abuse of notation here), and $w^{(k)}$ be the parameter obtained by $k$ iterations of the inner loop.

**Lemma D.10** (Existence of $W^*$). *Let $\Delta v := \sigma(w^{(0)} x_i) - V_i$. Then, there exists a $w^*$ such that $F_w(w^*) = 0$ and*

$$\left\| w^{(0)} - w^* \right\| \leq \frac{1}{\alpha s} \|\Delta v\|.$$

*Proof.* The proof is essentially same as that of Lemma D.7. $\qquad \square$

We then provide the convergence analysis to the update (12) by using the observation in Appendix D.2.

**Lemma D.11** (Convergence analysis of $W_1$). *Under the same condition as Theorem 5.1,*

$$\left\| \sigma\left( w^{(k)} X \right) - V_1 \right\|^2 \leq \frac{\ell^2 \cdot \max_i \|x_i\|^2}{\alpha^2 s^2} \exp\left( -\alpha^2 s^2 \gamma \eta_W^{(2)} k \right) \left\| \sigma\left( w^{(0)} X \right) - V_1 \right\|^2.$$

*Proof.* By letting $a \to i$, $b \to i$, $x_a \to x_i$ and $y_a \to V_i$ in Appendix D.2, we obtain

$$\left\| w^{(k+1)} - w^* \right\|^2 \tag{13}$$

$$= \left\| w^{(k)} - w^* \right\|^2 - 2\gamma \eta_W^{(2)} \nabla_w F_W \left( w^{(k)} \right)^\top \left( w^{(k)} - w^* \right) + \gamma^2 \eta_W^{(2)^2} \left\| \nabla_w F_W \left( w^{(k)} \right) \right\|^2.$$

The second term can be bounded by
$$\nabla_w F_w\left(w^{(k)}\right)^\top \left(w^{(k)} - w^*\right) \geq 2\lambda_{\min}\left(X^\top \Xi DX\right)\left\|w^{(k)} - w^*\right\|^2.$$
Then, we have
$$\lambda_{\min}\left(X^\top \Xi DX\right) \geq \alpha^2 \cdot \lambda_{\min}(X^\top X) = \alpha^2 s^2.$$
Moreover, the third term can be bounded by
$$\left\|\nabla_w F_W\left(w^{(k)}\right)\right\|^2 \leq 4\lambda_{\max}(DX^\top XD)\|\sigma(Xw) - \sigma(Xw^*)\|^2$$
$$\leq 4\ell^2 \lambda_{\max}(X^\top X)\left\|\sigma(Xw^{(k)}) - \sigma(Xw^*)\right\|^2$$
$$\leq \ell^2 \cdot \ell^2 \|X\|_{op}^2\left\|w^{(k)} - w^*\right\|^2 \leq \max_i\|x_i\|^2 \cdot \ell^4\left\|V^{(k)} - v^*\right\|^2,$$
where we use Lemma D.4 in the third inequality and Lemma D.1 in the third inequality.

By substituting these bounds to (13), we obtain
$$\left\|w^{(k+1)} - w^*\right\|^2$$
$$\leq \left\|w^{(k)} - w^*\right\|^2 - 2\alpha^2 s^2 \gamma \eta_W^{(2)}\left\|w^{(k)} - w^*\right\|^2 + \gamma^2 \eta_W^{(2)2}\ell^4 \max_i\|x_i\|^2\left\|w^{(k)} - w^*\right\|^2$$
$$\leq \left\|w^{(k)} - w^*\right\|^2 - \gamma \eta_W^{(2)}\left(2\alpha^2 s^2 - \gamma \eta_W^{(2)}\ell^4 \cdot \max_i\|x_i\|^2\right)\left\|w^{(k)} - w^*\right\|^2$$
$$\leq \left\|w^{(k)} - w^*\right\|^2 - \alpha^2 s^2 \gamma \eta_W^{(2)}\left\|w^{(k)} - w^*\right\|^2 = \left(1 - \alpha^2 s^2 \gamma \eta_W^{(2)}\right)\left\|w^{(k)} - w^*\right\|^2,$$
where we use $\eta_W^{(2)} \leq \frac{1}{\gamma \ell^4 \cdot \max_i\|x_i\|^2}$ in the last inequality. This implies
$$\left\|w^{(k)} - w^*\right\|^2 \leq \left(1 - \alpha^2 s^2 \eta_W^{(2)}\right)^k\left\|w^{(0)} - w^*\right\|^2 \leq \exp\left(-\alpha^2 s^2 \eta_W^{(2)} k\right)\left\|w^{(0)} - w^*\right\|^2,$$
where we use $1 - x \leq \exp(-x)$ in the first inequality, and hence,
$$\left\|\sigma(w^{(k)}X) - V_1\right\|^2 \leq \ell^2\left\|\left(w^{(k)} - w^*\right)X\right\|^2$$
$$\leq \ell^2 \cdot \max_i\|x_i\|^2\left\|w^{(k)} - w^*\right\|^2$$
$$\leq \ell^2 \cdot \max_i\|x_i\|^2 \exp\left(-\alpha^2 s^2 \eta_W^{(2)} k\right)\left\|w^{(0)} - w^*\right\|^2$$
$$\leq \frac{\ell^2 \cdot \max_i\|x_i\|^2}{\alpha^2 s^2} \exp\left(-\alpha^2 s^2 \eta_W^{(2)} k\right)\left\|\sigma\left(w^{(0)}X\right) - V_1\right\|^2,$$
which gives the conclusion. $\qquad\square$

### D.3.1  Proof of Theorem 5.1

Before providing the proof of Theorem 5.1, we introduce the following lemma:

**Lemma D.12** (Bound on $\Delta v$ at the output layer). *Let* $R_i := \left|W_j^{(0)}V_{L-1,i}^{(0)} - y_i\right|$. *Then, we have*
$$\left\|V_{L-1,i}^{(k)} - V_{L-1,i}^{(k-1)}\right\| \leq 4R_i\eta_V.$$

*Proof.* By the construction of the algorithm, we have
$$\left\|V_{L-1,i}^{(k)} - V_{L-1,i}^{(k-1)}\right\| = \left\|2\eta_V(W_L^{(k)}V_{L-1,i}^{(k-1)} - y_i)W_L^{(k)}\right\|$$
$$\leq 2\eta_V\left\|W_L^{(k)}\right\|_{op} \cdot \left\|W_L^{(k)}V_{L-1,i}^{(k-1)} - y_i\right\|$$
$$\leq 4\eta_V\left\|W_L^{(0)}V_{L-1,i}^{(0)} - y_i\right\| = 4\eta_V R_i,$$
which gives the conclusion. $\qquad\square$

Then, we move to the proof to Theorem 5.1.

*Proof of Theorem 5.1.* Let us consider the decomposition of $F$ as

$$F = F_L + \sum_{j=1}^{L-1} F_j = \sum_{i=1}^{n} \left[ F_{L,i} + \sum_{j=1}^{L-1} \sum_{p=1}^{r} F_{j,i,p} \right],$$

where

$$F_{L,i} := (W_L V_{L-1,i} - y_i)^2, \qquad F_L = \sum_{i=1}^{n} F_{L,i}$$

and

$$F_{j,i,p} := \gamma \Big( \sigma(W_j V_{j-1,i})_p - (V_{j,i})_p \Big)^2, \qquad F_j = \sum_{i=1}^{n} \sum_{p=1}^{r} F_{j,i,p}$$

for $j = 1, \ldots, L-1$. The proof consists of two parts: (I) $F_L$ is monotonically decreasing in the outer loop and (II) $F_{j,i,p}$ ($j = 1, \ldots, L-1, i = 1, \ldots, n, p = 1, \ldots, r$) is sufficiently small at the end of each inner iteration.

**(I) Bound on $F_L$** The update of $V_{L-1,i}$ is described by

$$V_{L-1,i}^{(k)} = V_{L-1,i}^{(k-1)} - 2\eta_V \left( W_L^{(k)} V_{L-1,i}^{(k-1)} - y_i \right) W_L^{(k)}.$$

Then, we have

$$W_L^{(k)} V_{L-1,i}^{(k-1)} - y_i = \left( 1 - 2\eta_V \left\| W_L^{(k)} \right\|^2 \right) \left( W_L^{(k)} V_{L-1,i}^{(k-1)} - y_i \right).$$

This results in

$$F_{L,i}^{(k)} \le \left( 1 - 2\eta_V \left\| W_L^{(k)} \right\|^2 \right)^2 F_{L,i}^{(k-1)} \le \exp \left( -4\eta_V \left\| W_L^{(k)} \right\|^2 \right) F_{L,i}^{(k-1)} \le \exp \left( -\eta_V \right) F_{L,i}^{(k-1)},$$

where the second inequality follows from $1 - x \le e^{-x}$ and the last inequality from $\left\| W_L^{(k)} \right\| \ge \frac{1}{2}$. This concludes

$$F_L^{(k)} \le \exp \left( -\eta_V k \right) F_L^{(0)}.$$

Since $F_L^{(0)} = R$ by the definition of $R$, after $k = \frac{1}{\eta_V} \log \left( \frac{3R}{\epsilon} \right)$ iterations, $F_L^{(k)} \le \frac{\epsilon}{3}$ holds.

**(II)-(i) Bound on $F_j$ ($j = 2, \ldots, L-1$)** Let us define $\Delta v_{j,i}^{(k)}$ as the initial value of $\sigma(W_{j+1} V_{j,i}) - V_{j+1,i}$ for $j = 1, \ldots, L-1$ when we update $V_j, i$ at the $k$th outer iteration, where we denote $V_{L,i} := y_i$. Then, by Lemma D.7 and Lemma D.9, we have

$$\left\| \Delta v_{j,i}^{(k)} \right\| \le \frac{2}{\alpha} \left( \left\| \Delta v_{j+1,i}^{(k)} \right\| + \sqrt{\epsilon} \right)$$

for any $j = 1, \ldots, L-2$ and $i = 1, \ldots, n$. We have $\left\| \Delta v_{L-1,i}^{(k)} \right\| \le 4R_{\max}\eta_V$ by Lemma D.12. By using this bound, we can derive

$$\left\| \Delta v_{j,i}^{(k)} \right\| \le \left( 4R_{\max}\eta_V + \frac{2}{2-\alpha}\sqrt{\epsilon} \right) \left( \frac{2}{\alpha} \right)^{L-1-j} - \frac{2}{2-\alpha}\sqrt{\epsilon} \tag{14}$$

$$\le \left( 4R_{\max}\eta_V + \frac{2}{2-\alpha}\sqrt{\epsilon} \right) \left( \frac{2}{\alpha} \right)^{L} \tag{15}$$

by induction. Indeed, (14) holds for $j = L - 1$ with equality. Moreover, under the induction hypothesis, it holds that

$$\left\|\Delta v_{j-1,i}^{(k)}\right\| \leq \frac{2}{\alpha}\left(\left\|\Delta v_{j,i}^{(k)}\right\| + \sqrt{\epsilon}\right)$$

$$\leq \frac{2}{\alpha}\left(\left(4R_{\max}\eta_V + \frac{2}{2-\alpha}\sqrt{\epsilon}\right)\left(\frac{2}{\alpha}\right)^{L-j} - \frac{2}{2-\alpha}\sqrt{\epsilon} + \sqrt{\epsilon}\right)$$

$$= \left(4R_{\max}\eta_V + \frac{2}{2-\alpha}\sqrt{\epsilon}\right)\left(\frac{2}{\alpha}\right)^{L-(j-1)} - \frac{2}{2-\alpha}\sqrt{\epsilon}.$$

This concludes (14) for $j = 1, \ldots, L - 1$. Then, by using Lemma D.8, we have

$$F_{j,i,p} \leq \gamma \cdot \frac{16\ell^2}{\alpha^2}\exp\left(-\frac{\alpha^2}{4}\gamma\eta_V k\right)\left\|\sigma\left(WV^{(0)}\right) - V'\right\|^2 \cdot \left(\frac{2}{\alpha}\right)^L\left(4R_{\max}\eta_V + \frac{2}{2-\alpha}\sqrt{\epsilon}\right)^2.$$

Thus,

$$k_{in} = \frac{4}{\gamma\alpha^2\ell\eta_V}\log\left(\left(\frac{2}{\alpha}\right)^L\left(4R_{\max}\eta_V + \frac{2}{2-\alpha}\sqrt{\epsilon}\right)^2\frac{48\ell^2(L-2)rn\gamma}{\alpha^2\epsilon}\right)$$

gives $F_{j,i,p} \leq \frac{\epsilon}{3(L-2)rn}$ and hence, $F_j \leq \frac{\epsilon}{3(L-2)}$ by summing up $F_{j,i,p}$.

**(II)-(ii) Bound on $F_1$**    By using Lemma D.11, we have

$$\sum_{i=1}^n F_{1,i,p} \leq \frac{\ell^2 \cdot \max_i\|x_i\|^2}{\alpha^2 s^2}\exp\left(-\alpha^2 s^2\gamma\eta_W^{(2)}k_{in}\right)\left\|\sigma(W^{(0)}U) - V_1\right\|^2$$

Since $\Delta v_{1,i}^{(k)} \leq \left(4R_{\max}\eta_V + \frac{2}{2-\alpha}\sqrt{\epsilon}\right)\left(\frac{2}{\alpha}\right)^{L-1}$, we have

$$\left\|\sigma(W^{(0)}U) - V_1\right\|^2 \leq \sum_{i=1}^n\left(\frac{2}{\alpha}(\|\Delta v_{1,i}\| + \epsilon)\right)^2$$

$$= n\left(4R_{\max}\eta_V + \frac{2}{2-\alpha}\sqrt{\epsilon}\right)^2 \cdot \left(\frac{2}{\alpha}\right)^{2L}$$

Thus,

$$k_{in} = \frac{1}{\alpha^2 s^2\gamma\eta_W^{(2)}}\log\left(n\left(R_{\max}\eta_V + \frac{2}{2-\alpha}\sqrt{\epsilon}\right)^2\left(\frac{2}{\alpha}\right)^{2L}\cdot\frac{3\ell^2 \cdot \max_i\|x_i\|^2 r}{\alpha^2 s^2\epsilon}\right)$$

gives $\sum_{i=1}^n F_{1,i,p} \leq \frac{\epsilon}{r}$ for $p = 1, \ldots, r$. This results in $F_1 = \sum_{i=1}^n\sum_{p=1}^r F_{1,i,p} \leq \frac{\epsilon}{3}$.

**(III) Summing up all**    By combining all, after $K$ outer iterations and $K_V$ and $K_W$ inner iterations, we have

$$F = F_L + \sum_{j=1}^{L-1} F_j \leq \underbrace{\frac{\epsilon}{3}}_{F_L} + \sum_{j=2}^{L-1}\underbrace{\frac{\epsilon}{3(L-2)}}_{F_2\ldots,F_{L-1}} + \underbrace{\frac{\epsilon}{3}}_{F_1} = \epsilon,$$

which gives the conclusion.    □

## E    Proof of Theorem 5.5

Here, we provide the proof of Theorem 5.5, the generalization error bound of neural networks trained by Algorithm 1.

*Proof of Theorem 5.5.* By using the bound on $u$ and $y$ supposed in Assumption 5.3, we have

$$
\begin{aligned}
|f(u) - y| &\leq B_Y + |W_L \sigma(W_{L-1} \ldots \sigma(W_1 u) \ldots)| \\
&\leq B_Y + \ell \|W_L\|_{op} \|W_{L-1} \sigma(W_{L-2} \ldots \sigma(W_1 u) \ldots)\| \\
&\leq \ldots \\
&\leq B_Y + \ell^{L-1} \left( \prod_{j=2}^{L} \|W_j\|_{op} \right) \|W_1 u\| \\
&\leq B_Y + 2^L \ell^{L-1} \|u\| \leq B_Y + 2^L \ell^{L-1} B_X.
\end{aligned}
$$

Hence, by taking $M = B_Y + 2^L \ell^{L-1} B_X$ and $\mathcal{R}(\mathcal{F})$ as what derived by Lemma E.2 in Lemma E.1, we obtain the conclusion. □

**Lemma E.1** (Theorem 11.3 in [23]). *For a hypothesis class $\mathcal{F}$ and a training data $\{(x_i, y_i)\}_{i=1}^n$, let us define its (empirical) Rademacher complexity by*

$$
\mathcal{R}(\mathcal{F}) := \mathbb{E}_\sigma \left[ \sup_{f \in \mathcal{F}} \frac{\sigma^\top f(\mathbf{u})}{n} \right],
$$

*where $f(\mathbf{x}) = (f(x_1), \ldots, f(x_n))^\top$ and $\sigma$ is a random vector whose each component independently takes value $\pm 1$ with probability $\frac{1}{2}$. Suppose that $|h(x) - y| \leq M$ a.s. for any $h \in \mathcal{F}$. Then, for any $0 < \delta < 1$, with probability at least $1 - \delta$ over a sample, we have*

$$
\mathbb{E}_{(x,y) \sim P} \left[ (h(x) - y)^2 \right] \leq \frac{1}{n} \sum_{i=1}^n (h(x_i) - y_i)^2 + 2M\mathcal{R}(\mathcal{F}) + 3M^2 \sqrt{\frac{\log(2/\delta)}{2n}}.
$$

**Lemma E.2** (Rademacher complexity bound). *Let $\mathcal{F}$ be the class of neural network predictors obtained by Algorithm 1. Then, the Rademacher complexity of $\mathcal{F}$ can be bounded by*

$$
\mathcal{R}(\mathcal{F}) \leq \frac{4}{n\sqrt{n}} + \log \left( \frac{1}{\sqrt{n}} \right) \frac{12\sqrt{R_\mathcal{F}}}{n}
$$

*with $R_\mathcal{F} = d_{in}(2r)^L L^3 \|U\|^2 \log(2r^2)(\log n)$.*

To obtain this result, we apply the obtained bound on the spectral of $W$ to the Rademacher complexity bound shown in [8] as follows:

**Lemma E.3** (Lemma A.8 in [8]). *Assume activation functions $\{\sigma_j(\cdot)\}_{j=1}^L$ such that each $\sigma_j$ is $\rho_j$-Lipschitz continuous and $\sigma_j(0) = 0$. Let us define*

$$
\mathcal{F} := \left\{ \sigma_L(W_L \sigma_{L-1}(\ldots \sigma_1(W_1 \cdot) \ldots)) \mid \|W_j\|_{op} \leq B_j, \; \|W_j\|_{2,1} \leq b_j \; (1 \leq j \leq L) \right\}.
$$

*Then, it holds that*

$$
\mathcal{R}(\mathcal{F}) \leq \frac{4}{n\sqrt{n}} + \log \left( \frac{1}{\sqrt{n}} \right) \frac{12\sqrt{R_\mathcal{F}}}{n},
$$

*where $R_\mathcal{F} > 0$ is a constant defined by*

$$
R_\mathcal{F} := \|X\|^2 \log(2r^2)(\log n) \left( \prod_{j=1}^L B_j \rho_j \right) \left( \sum_{j=1}^L \left( \frac{b_j}{B_j} \right)^{\frac{2}{3}} \right)^3.
$$

*Proof of Lemma E.2.* By applying Lemma E.3 with $\rho_1 = \cdots = \rho_L = 1$, $B_j = 2$, $b_1 = 2d_{in}$ and $b_j = 2r$ for $j = 1, \ldots, L-2$ and $b_L = 2$, we obtain

$$
R_\mathcal{F} = \|X\|^2 \log(2r^2)(\log n) \left( 4d_{in} \prod_{j=2}^{L-1} (2r) \right) \left( \sum_{j=1}^L \left( \frac{2r}{2} \right)^{\frac{2}{3}} \right)^3
$$

$$
= \|X\|^2 \log(2r^2)(\log n) \cdot 4d_{in}(2r)^{L-2} L^3 r^2 = d_{in}(2r)^L L^3 \|U\|^2 \log(2r^2)(\log n),
$$

which gives the conclusion. □

# F Proof of Theorem 6.3

## F.1 Proof of Lemma 6.2

*Proof of Lemma 6.2.* First, we have $\mathbb{E}[\|w_+\|^2] = \mathbb{E}[\|w_-\|^2] = \frac{1}{2}$. The first equality follows from the symmetricity, and the second equality follows from

$$\frac{1}{2} = \frac{1}{2}\mathbb{E}\left[\|W_L\|^2\right] = \frac{1}{2}\mathbb{E}\left[\|w_+\|^2 + \|w_-\|^2\right] = \frac{1}{2}\left(\mathbb{E}\left[\|w_+\|^2\right] + \mathbb{E}\left[\|w_-\|^2\right]\right) = \mathbb{E}[\|w_+\|^2],$$

where we use $\mathbb{E}[\|w_+\|^2] = \mathbb{E}[\|w_-\|^2]$ in the last equality. Then, by using the concentration inequality argument (see Example 2.11 in [36] for example), we have

$$\mathrm{P}\left(\left|\|w_+\|^2 - \frac{1}{2}\right| \geq t\right) \leq 2\exp\left(-\frac{rt^2}{8}\right)$$

for any $t \in (0, 1)$. By letting $t = \sqrt{\frac{8\log(2/\delta)}{r}}$, we obtain

$$\mathrm{P}\left(\|w_+\|^2 < \frac{1}{2} - \sqrt{\frac{8\log(2/\delta)}{r}}\right) \leq \delta$$

Since the same argument holds with $w_-$, taking a union bound concludes the assertion. $\square$

## F.2 Analysis of gradient descent with skip connection

We introduce the key idea of analysis with general notations similarly to Appendix D, while there exists a skip connection. Let us consider the regression problem with an objective

$$F_{relu}(w) := \sum_{a=1}^{b}\left(\sigma\left(w^\top x_a\right) + w_a - y_a\right)^2,$$

where $w \in \mathbb{R}^d$ is a trainable parameter. Let $w' := w - \eta\nabla_w \sum_{a=1}^{b}\left(\sigma\left(w^\top x_a\right) + w_a - y_a\right)^2$, where $w'$ denotes the parameter obtained by a single update of gradient descent with a step-size $\eta > 0$. Denote $X := (x_1, \ldots x_b)^\top \in \mathbb{R}^{b \times d}$ and $Y := (y_1, \ldots, y_b)^\top \in \mathbb{R}^b$. Then, $\sum_{a=1}^{b}\left(\sigma(w^\top x_a) + w_a - y_a\right)^2 = \|\sigma(Xw) + w - Y\|^2$ holds and a straightforward calculation shows $w' = w - 2\eta(X^\top D + I)(\sigma(Xw) + w - Y)$, where $D = \mathrm{diag}\left((\sigma'(w^\top x_1), \ldots, \sigma'(w^\top x_b))\right)$.

Now, we assume that there exists a unique optimal solution $w^*$ satisfying $F_{relu}(w^*) = 0$, i.e., $Y = \sigma(Xw^*) + w^*$. Then, we have

$$\begin{aligned}
\|w' - w^*\|^2 &= \|w - \eta\nabla_w F_{relu}(w) - w^*\|^2 \\
&= \|w - w^*\|^2 - 2\eta\nabla_w F_{relu}(w)^\top(w - w^*) + \eta^2\|\nabla_w F_{relu}(w)\|^2
\end{aligned}$$

and

$$\begin{aligned}
&\nabla_w F_{relu}(w)^\top(w - w^*) \\
&= 2(\sigma(Xw) + w - \sigma(Xw^*) - w^*)^\top(DX + I)(w - w^*) \\
&= 2[(\Xi X + I)(w - w^*)]^\top DX(w - w^*) \\
&= 2(w - w^*)^\top(X^\top\Xi + I)(DX + I)(w - w^*) \\
&= 2\|w - w^*\|^2 + 2(w - w^*)^\top X^\top\Xi DX(w - w^*) + 2(w - w^*)^\top(X^\top\Xi + DX)(w - w^*),
\end{aligned} \tag{16}$$

where $\Xi$ is a diagonal matrix whose all diagonal components are within $[0, 1]$, whose existence is guaranteed by the same argument as Proposition D.3. Then, we evaluate the right hand side.

**Lemma F.1.** $O \preceq D \preceq I$ holds.

*Proof.* The assertion directly follows from $\sigma'(u) \in \{0, 1\}$ for arbitrary $u \in \mathbb{R}$. $\square$

**Lemma F.2.** *Suppose $\|X\|_{op} \leq \frac{1}{3}$. Then, the inequality (16) $\geq \frac{4}{3}\|w - w^*\|^2$ holds.*

*Proof.* Since $X^\top \Xi D X$ is a positive semi-definite matrix, we have

$$(16) \geq 2\|w - w^*\|^2 - 2\|X^\top \Xi + DX\|_{op}\|w - w^*\|^2$$
$$\geq \left(2 - 2 \cdot \frac{1}{3}\right)\|w - w^*\| \geq \frac{4}{3}\|w - w^*\|^2,$$

which gives the conclusion. $\qquad\square$

Besides the lower bound on (16), we have the upper bound of the gradient as

$$\|\nabla_w F_{relu}(w)\|^2 = \left\|2X^\top D(\sigma(Xw) + w - Y)\right\|^2 \tag{17}$$
$$\leq 4\lambda_{\max}((XD + I)^\top(XD + I))\|\sigma(Xw) + w - Y\|^2,$$

where

$$\lambda_{\max}((XD + I)^\top(XD + I)) \geq 0$$

is the largest eigenvalue of the matrix $(XD + I)^\top(XD + I)$.

Moreover, we provide several lemmas, which we utilize in the proof of Theorem 6.3.

**Lemma F.3.** *Suppose $\|W\|_{op} \leq \frac{1}{3}$ and $V' \geq 0$. Then, if $\|\sigma(WV) + V - V'\|^2 \leq \epsilon$, then*

$$\left\|V - (V)^+\right\|^2 \leq \epsilon, \qquad \left\|\sigma\big(W(V)^+\big) + (V)^+ - V'\right\|^2 \leq \frac{49}{9}\epsilon.$$

*Proof.* Since $\sigma(WV) \geq 0$ and $V \geq 0$, we have

$$\epsilon \geq \|\sigma(WV) + V - V'\|^2 \geq \sum_{V_j < 0}\left[\sigma(WV)_j + V_j - V_j'\right]^2 \geq \sum_{V_j < 0}(V_j)^2 = \left\|V - (V)^+\right\|^2,$$

which gives the first conclusion. The second follows from

$$\left\|\sigma\big(W(V)^+\big) + (V)^+ - V'\right\|$$
$$\leq \left\|\sigma\big(W(V)^+\big) - \sigma(WV) + (V)^+ - V\right\| + \left\|\sigma(WV) + V - V'\right\|$$
$$\leq \left\|\sigma\big(W(V)^+\big) - \sigma(WV)\right\| + \left\|(V)^+ - V\right\| + \left\|\sigma(WV) + V - V'\right\|$$
$$\leq \left\|W\big((V)^+ - V\big)\right\| + \epsilon^{\frac{1}{2}} + \epsilon^{\frac{1}{2}} \leq \frac{1}{3}\epsilon^{\frac{1}{2}} + 2\epsilon^{\frac{1}{2}} = \frac{7}{3}\epsilon^{\frac{1}{2}},$$

where we use the triangle inequality in the first and second inequalities, and 1-Lipschitzness of the ReLU activation in the third inequality. $\qquad\square$

**Lemma F.4.** *Suppose that $V^{(0)}$ satisfies $\sigma(WV^{(0)}) + V^{(0)} - V' =: \Delta v$ and $V^*$ satisfies $\sigma(WV^*) + V^* = V'$. If $\|W\|_{op} < 1$, it holds that*

$$\left\|V^{(0)} - V^*\right\| \leq \frac{1}{1 - \|W\|_{op}}\|\Delta v\|.$$

*Proof.* We have

$$\|\Delta v\| = \left\|\sigma(WV^{(0)}) + V^{(0)} - V'\right\| = \left\|\sigma(WV^{(0)}) + V^{(0)} - \sigma(WV^*) - V^*\right\|$$
$$\geq \left\|V^{(0)} - V^*\right\| - \left\|\sigma(WV^{(0)}) - \sigma(WV^*)\right\|$$
$$\geq \left\|V^{(0)} - V^*\right\| - \left\|W(V^{(0)} - V^*)\right\|$$
$$\geq \left\|V^{(0)} - V^*\right\| - \|W\|_{op}\left\|V^{(0)} - V^*\right\|$$
$$= \left(1 - \|W\|_{op}\right)\left\|V^{(0)} - V^*\right\|,$$

where we use the the triangle inequality in the first inequality, the 1-Lipschitzn continuity of ReLU activation in the second inequality. Dividing each term by $1 - \|W\|_{op}$ gives the conclusion. $\qquad\square$

### F.3 Preliminary Results

**Lemma F.5** (Regularity of weight matrix $W_j$ during training). *For $j = 2, \ldots, L - 1$, $\|W_j\|_{op} \leq \frac{1}{3}$ always holds during the training.*

*Proof.* By Lemma D.5, it suffices to show that every of $W_j$ satisfies $\|\Delta w\| \leq \frac{1}{12\sqrt{r}}$, where $\Delta w$ denotes the difference between $w$ at the start and end of the training by the same as the proof of Lemma D.4.

To this end, we prove $\|\Delta w\| \leq \frac{1}{12\sqrt{r}}$. This follows from

$$
\eta_W^{(1)} \gamma \nabla_w \|\sigma(wV) + V - V'\|^2 = 2\eta_W^{(1)} \gamma \cdot \left\| \mathrm{diag}(\sigma'(wV)) V^\top (\sigma(wV) + V - V') \right\|
$$
$$
\leq 2\eta_W^{(1)} \gamma \ell \lambda_{\max}^{1/2}(VV^\top) \cdot \|\sigma(wV) + V - V'\|
$$
$$
\leq 2\eta_W^{(1)} \gamma \ell C_V \cdot \eta_V \left( \frac{3}{2} \right)^L \leq \frac{1}{12K\sqrt{r}},
$$

where the last inequality follows from the definition of $\eta_W^{(1)}$. $\qquad\square$

**Lemma F.6** (Bound on $\Delta v$ at the output layer). *Let $R_i := \left| W_j^{(0)} V_{L-1,i}^{(0)} - y_i \right|$. Then, we have*

$$
\left\| V_{L-1,i}^{(k)} - V_{L-1,i}^{(k-1)} \right\| \leq 4R_i \eta_V.
$$

*Proof.* Since $V_{L-1}^{(k)} \geq 0$, we have

$$
\left\| V_{L-1,i}^{(k)} - V_{L-1,i}^{(k-1)} \right\| = \left\| \left( V_{L-1,i}^{(k-1)} - 2\eta_V \left( W_L V_{L-1,i}^{(k-1)} - y_i \right) W_L \right)^+ - V^{(k-1)} \right\|
$$
$$
\leq \left\| \left( V_{L-1,i}^{(k-1)} - 2\eta_V \left( W_L V_{L-1,i}^{(k-1)} - y_i \right) W_L \right) - V^{(k-1)} \right\|
$$
$$
= \left\| 2\eta_V (W_L^{(k)} V_{L-1,i}^{(k-1)} - y_i) W_L^{(k)} \right\|
$$
$$
\leq 2\eta_V \left\| W_L^{(k)} \right\|_{op} \cdot \left\| W_L^{(k)} V_{L-1,i}^{(k-1)} - y_i \right\|
$$
$$
\leq 4\eta_V \left\| W_L^{(0)} V_{L-1,i}^{(0)} - y_i \right\| = 4\eta_V R_i,
$$

which gives the conclusion. $\qquad\square$

### F.4 Proof of Theorem 6.3

*Proof of Theorem 6.3.* We follow the similar argument as that of Theorem 5.1. Let us consider the decomposition of $F$ as

$$
F = F_L + \gamma \sum_{j=1}^{L-1} F_j = \sum_{i=1}^{n} \left[ F_{L,i} + \gamma \sum_{j=1}^{L-1} \sum_{p=1}^{r} F_{j,i,p} \right],
$$

where

$$
F_{L,i} := (W_L V_{L-1,i} - y_i)^2, \qquad F_L = \sum_{i=1}^{n} F_{L,i}
$$

and

$$
F_{j,i,p} := \left( \sigma(W_j V_{j-1,i})_p + (V_{j-1,i})_p - (V_{j,i})_p \right)^2, \qquad F_j = \sum_{i=1}^{n} \sum_{p=1}^{r} F_{j,i,p}
$$

for $j = 1, \ldots, L - 1$.

**(I) Bound on $F_L$**  We only need to consider the case $W_L V_{L-1,i}^{(k-1)} - y_i \neq 0$. The update of $V_{L-1,i}$ is described by

$$V_{L-1,i}^{(k)} = \left(V_{L-1,i}^{(k-1)} - 2\eta_V\left(W_L V_{L-1,i}^{(k-1)} - y_i\right)W_L\right)^+ = V_{L-1,i}^{(k)} - 2\eta_V(W_L V_{L-1,i}^{(k-1)} - y_i)\tilde{w},$$

where we define $\tilde{w} := (2\eta_V(W_L V_{L-1,i}^{(k-1)} - y_i))^{-1}\left(V_{L-1,i}^{(k-1)} - V_{L-1,i}^{(k)}\right)$. Then, we have

$$W_L^{(k)} V_{L-1,i}^{(k-1)} - y_i = \left(1 - 2\eta_V \tilde{w}^\top W_L\right)\left(W_L V_{L-1,i}^{(k-1)} - y_i\right).$$

Then, we show an inequality

$$\tilde{w}^\top W_L \geq \min\{\|w_+\|^2, \|w_-\|^2\}. \tag{18}$$

First we consider a case $W_L V_{L-1,i}^{(k-1)} - y_i > 0$. In this case, we have

$$\left(2\eta_V\left(W_L V_{L-1,i}^{(k-1)} - y_i\right)\tilde{w}\right)_j$$

$$= \left(\left(V_{L-1,i}^{(k-1)} - 2\eta_V\left(W_L V_{L-1,i}^{(k-1)} - y_i\right)W_L\right)^+ - V_{L-1,i}^{(k-1)}\right)_j$$

$$= \begin{cases} 2\eta_V\left(W_L V_{L-1,i}^{(k-1)} - y_i\right)(W_L)_j & \text{if } j \in J_1 := \left\{j \mid (W_L)_j \leq 0\right\}, \\ 2\eta_V\left(W_L V_{L-1,i}^{(k-1)} - y_i\right)(W_L)_j & \text{if } j \in J_2 := \left\{j \mid (W_L)_j > 0 \text{ and } V_{L-1,i}^{(k-1)} > 2\eta_V\left(W_L V_{L-1,i}^{(k-1)} - y_i\right)(W_L)_j\right\}, \\ \left(V_{L-1,i}^{(k-1)}\right)_j & \text{otherwise.} \end{cases}$$

This gives

$$\tilde{w}^\top W_L = \sum_{j=1}^r (\tilde{w})_j (W_L)_j$$

$$= \sum_{j \in J_1 \cup J_2} (W_L)_j^2 + \sum_{j \in (J_1 \cup J_2)^c} 2\eta_V\left(W_L V_{L-1,i}^{(k-1)} - y_i\right)^{-1}\left(V_{L-1,i}^{(k-1)}\right)_j (W_L)_j$$

$$\geq \sum_{j \in J_1} (W_L)_j^2 = \|w_-\|^2, \tag{19}$$

where in the inequality we use $\left(V_{L-1,i}^{(k-1)}\right)_j > 0$ and $(W_L)_j > 0$ for $j \in (J_1 \cup J_2)^c$.

If $W_L V_{L-1,i}^{(k-1)} - y_i < 0$, it holds that

$$\left(2\eta_V\left(W_L V_{L-1,i}^{(k-1)} - y_i\right)\tilde{w}\right)_j$$

$$= \begin{cases} 2\eta_V\left(W_L V_{L-1,i}^{(k-1)} - y_i\right)(W_L)_j & \text{if } j \in J_1 := \left\{j \mid (W_L)_j \geq 0\right\}, \\ 2\eta_V\left(W_L V_{L-1,i}^{(k-1)} - y_i\right)(W_L)_j & \text{if } j \in J_2 := \left\{j \mid (W_L)_j < 0 \text{ and } V_{L-1,i}^{(k-1)} > 2\eta_V\left(W_L V_{L-1,i}^{(k-1)} - y_i\right)(W_L)_j\right\}, \\ \left(V_{L-1,i}^{(k-1)}\right)_j & \text{otherwise.} \end{cases}$$

This gives

$$\tilde{w}^\top W_L = \sum_{j=1}^r (\tilde{w})_j (W_L)_j$$

$$= \sum_{j \in J_1 \cup J_2} (W_L)_j^2 + \sum_{j \in (J_1 \cup J_2)^c} 2\eta_V\left(W_L V_{L-1,i}^{(k-1)} - y_i\right)^{-1}\left(V_{L-1,i}^{(k-1)}\right)_j (W_L)_j$$

$$\geq \sum_{j \in J_1} (W_L)_j^2 = \|w_+\|^2, \tag{20}$$

where in the inequality we use $\left(V_{L-1,i}^{(k-1)}\right)_j > 0$ and $(W_L)_j < 0$ for $j \in (J_1 \cup J_2)^c$. The two bounds (19) and (20) conclude (18).

This results in

$$
\begin{aligned}
F_{L,i}^{(k)} &\leq \left(1 - 2\eta_V \tilde{w}^\top W_L\right)^2 F_{L,i}^{(k-1)} \\
&\leq \exp\left(-4\eta_V \tilde{w}^\top W_L\right) F_{L,i}^{(k-1)} \leq \exp\left(-4\eta_V \min\left\{\|w_+\|^2, \|w_-\|^2\right\}\right) F_{L,i}^{(k-1)},
\end{aligned}
$$

where the second inequality follows from $1 - x \leq e^{-x}$ and the last inequality from (18). This concludes

$$
F_L^{(k)} \leq \exp\left(-4\eta_V \min\left\{\|w_+\|^2, \|w_-\|^2\right\}k\right) F_L^{(0)}.
$$

Since $F_L^{(0)} = R$ by the definition of $R$, as long as we set $\eta_V \leq \frac{1}{2\min\{\|w_+\|^2, \|w_-\|^2\}}$ after $k = \frac{1}{4\eta_V \min\{\|w_+\|^2, \|w_-\|^2\}} \log\left(\frac{3R}{\epsilon}\right)$ iterations, $F_L^{(k)} \leq \frac{\epsilon}{3}$ holds.

**(II)-(i) Bound on $F_j$ $(j = 2, \dots, L-1)$** Let us define $\Delta v_{j,i}^{(k)}$ as the initial value of $\sigma(W_{j+1}V_{j,i}) + V_{j,i} - V_{j+1,i}$ for $j = 1, \dots, L-1$ when we update $V_{j,i}$ at the $k$th outer iteration, where we denote $V_{L,i} := y_i$. Then, $\left\|\Delta v_{j,i}^{(k)}\right\| \leq 2\eta_V R_i$ holds and Lemma F.4 gives

$$
\left\|\Delta v_{j,i}^{(k)}\right\| \leq \frac{1}{1 - \|W_{j+1}\|}\left(\left\|\Delta v_{j+1,i}^{(k)}\right\| + \sqrt{\epsilon}\right) + \epsilon \leq \frac{3}{2}\left(\left\|\Delta v_{j+1,i}^{(k)}\right\| + \frac{5}{3}\sqrt{\epsilon}\right).
$$

By these inequalities, we can ensure

$$
\left\|\Delta v_{j,i}^{(k)}\right\| \leq \left(4R_{\max}\eta_V + 5\sqrt{\epsilon}\right)\left(\frac{3}{2}\right)^L
$$

by taking $\alpha = \frac{4}{3}$ in (14).

By the observation in Appendix F.2, for each $j$, let $v^* \in \mathbb{R}^r$ be a solution of $\sigma(W_{j+1}v^*) + v^* - V_{j+1,i} = 0$. Let $\{V^{(k_{in})}\}_{k_{in}}$ be a sequence generated by the gradient descent. Then, it holds that

$$
\begin{aligned}
&\left\|V^{(k_{in}+1)} - v^*\right\|^2 \\
&= \left\|V^{(k_{in})} - v^*\right\|^2 - 2\gamma\eta_V \nabla_V F_{j,i}^\top\left(V^{(k_{in})} - v^*\right) + \gamma^2\eta_V^2\left\|\nabla_V F_{j,i}(V^{(k_{in})})\right\|^2.
\end{aligned}
\tag{21}
$$

For the second term, Lemma F.2 gives

$$
\nabla_V F_{j,i}^\top\left(V^{(k_{in})} - v^*\right) \geq \frac{4}{3}\left\|V^{(k_{in})} - v^*\right\|^2.
$$

For the third term, (17) gives

$$
\begin{aligned}
&\left\|\nabla_V F_{j,i}(V^{(k_{in})})\right\|^2 \\
&\leq 4\lambda_{\max}\left((W_{j+1}D + I)^\top(W_{j+1}D + 1)\right)\left\|\sigma(W_{j+1}V^{(k_{in})}) + V^{(k_{in})} - V_{j+i,i}\right\|^2 \\
&\leq 4 \cdot \left(\|W_{j+1}D\|_{op} + 1\right)^2\left\|\sigma(W_{j+1}V^{(k_{in})}) - \sigma(W_{j+1}v^*) + V^{(k_{in})} - v^*\right\|^2 \\
&\leq 4 \cdot \left(\frac{4}{3} + 1\right)^2 \cdot 2\left(\left\|\sigma(W_{j+1}V^{(k_{in})}) - \sigma(W_{j+1}v^*)\right\|^2 + \left\|V^{(k_{in})} - v^*\right\|^2\right) \\
&\leq \frac{128}{9}\left(\frac{1}{9} + 1\right)\left\|V^{(k_{in})} - v^*\right\|^2 = \frac{1280}{81}\left\|V^{(k_{in})} - v^*\right\|^2 \leq 16\left\|V^{(k_{in})} - v^*\right\|^2,
\end{aligned}
$$

where we use $(a + b)^2 \leq 2(a^2 + b^2)$ in the third inequality and $\|W_{j+1}\|_{op} \leq \frac{1}{3}$ in the third and fourth inequalities. Then, by substituting these bounds to (21), we obtain

$$\left\|V^{(k_{in}+1)} - v^*\right\|^2$$

$$\leq \left\|V^{(k_{in})} - v^*\right\|^2 - \frac{8}{3}\gamma\eta_V\left\|V^{(k_{in})} - v^*\right\|^2 + 16\gamma^2\eta_V^2\left\|V^{(k_{in})} - v^*\right\|^2$$

$$= \left\|V^{(k_{in})} - v^*\right\|^2 - \frac{8}{3}\gamma\eta_V(1 - 6\gamma\eta_V)\left\|V^{(k_{in})} - v^*\right\|^2$$

$$\leq \left\|V^{(k_{in})} - v^*\right\|^2 - \frac{4}{3}\gamma\eta_V\left\|V^{(k_{in})} - v^*\right\|^2 = \left(1 - \frac{4}{3}\gamma\eta_V\right)\left\|V^{(k_{in})} - v^*\right\|^2,$$

where we use $\eta_V \leq \frac{1}{12\gamma}$ in the second inequality.

This results in

$$\left\|V^{(k_{in})} - v^*\right\|^2 \leq \left(1 - \frac{4}{3}\gamma\eta_V\right)^{k_{in}}\left\|V^{(0)} - v^*\right\|^2 \leq \exp\left(-\frac{4}{3}\gamma\eta_V k_{in}\right)\left\|V^{(0)} - v^*\right\|^2,$$

where we use $1 - x \leq \exp(-x)$ in the last inequality, and hence,

$$\left\|\sigma\left(W_{j+1}V^{(k_{in})}\right) + V^{(k_{in})} - V_{j+1,i}\right\|^2 \leq 2\left(1 + \frac{1}{9}\right)\left\|V^{(k_{in})} - v^*\right\|^2$$

$$\leq \frac{20}{9}\exp\left(-\frac{4}{3}\gamma\eta_V k_{in}\right)\left\|V^{(0)} - v^*\right\|^2$$

$$\leq \frac{20}{9}\exp\left(-\frac{4}{3}\gamma\eta_V k_{in}\right) \cdot \frac{9}{4}\left\|\Delta v_{j,i}^{(k)}\right\|^2$$

$$\leq 5\exp\left(-\frac{4}{3}\gamma\eta_V k_{in}\right)\left(4R_{\max}\eta_V + 5\sqrt{\epsilon}\right)^2\left(\frac{3}{2}\right)^{2L},$$

where in the third inequality, we use $\left\|\Delta v_{j,i}^{(k)}\right\| \geq \frac{2}{3}\|V^{(0)} - v^*\|$, following from

$$\left\|\Delta v_{j,i}^{(k)}\right\| = \left\|\sigma\left(W_{j+1}V^{(0)}\right) + V^{(0)} - \sigma(W_{j+1}v^*) - v^*\right\|$$

$$\geq \left\|V^{(0)} - v^*\right\| - \left\|\sigma\left(W_{j+1}V^{(0)}\right) - \sigma(W_{j+1}v^*)\right\|$$

$$\geq \left\|V^{(0)} - v^*\right\| - \|W_{j+1}\|_{op}\left\|V^{(0)} - v^*\right\| \geq \frac{2}{3}\left\|V^{(0)} - v^*\right\|.$$

Hence, by taking $k_{in} = \frac{3}{4\gamma\eta_V}\log\left(\left(4R_{\max}\eta_V + 5\sqrt{\epsilon}\right)^2\left(\frac{3}{2}\right)^{2L}\frac{245(L-2)rn}{3\epsilon}\right)$, we obtain $F_{j,i} \leq \frac{3\epsilon}{49(L-2)rn}$. Then, Lemma F.3 gives $F_{j,i} \leq \frac{\epsilon}{3(L-2)rn}$ after the non-negative projection (line 10) is applied.

**(II)-(ii) Bound on $F_1$** The update of $W_1$ is same as what we considered in Theorem 5.1 (Algorithm 1) by setting $\alpha = \ell = 1$. Therefore, by using Lemma D.11, we have

$$F_1 \leq \exp\left(-s^2\eta_W^{(2)}k\right)\left\|\sigma\left(W_1^{(0)}X\right) - V_1\right\|^2$$

$$\leq \frac{\max_i\|x_i\|^2}{s^2}\exp\left(-s^2\eta_W^{(2)}k\right) \cdot \left(4R_{\max}\eta_V + 5\sqrt{\epsilon}\right)^2\left(\frac{3}{2}\right)^{2L}.$$

Thus, $k = \frac{1}{s^2\eta_W^{(2)}}\log\left(\left(4R_{\max}\eta_V + 5\sqrt{\epsilon}\right)\left(\frac{3}{2}\right)^L\frac{3\max_i\|x_i\|^2}{s^2\epsilon}\right)$ gives $F_1 \leq \frac{\epsilon}{3}$.

**(III) Summing up all** By combining all, after $K$ iterations and $K_V$ and $K_W$ iterations, we have

$$F = F_L + \sum_{j=1}^{L-1}F_j \leq \underbrace{\frac{\epsilon}{3}}_{F_L} + \sum_{j=2}^{L-1}\underbrace{\frac{\epsilon}{3\gamma(L-2)rn}rn\epsilon}_{F_2...,F_{L-1}} + \underbrace{\frac{\epsilon}{3}}_{F_1} = \epsilon,$$

which gives the conclusion. $\qquad\square$

# G Additional Experiments on Deeper Architectures

We conducted additional experiments to evaluate the scalability of the proposed BCD algorithm on deeper networks trained with the LeakyReLU activation ($\alpha = 0.5$). Networks with depths $L = 4, 8$, and 12 were trained on the same synthetic dataset and with the same hyperparameters as in Section 7.1.

Figure 4 illustrates the training loss trajectories for each setting. As expected, deeper architectures exhibit slower initial convergence due to increased optimization complexity. Nevertheless, the loss consistently decreases over epochs for all depths, demonstrating that the proposed method remains stable and effective even for substantially deeper models, in agreement with the theoretical convergence results presented in Theorem 5.1.

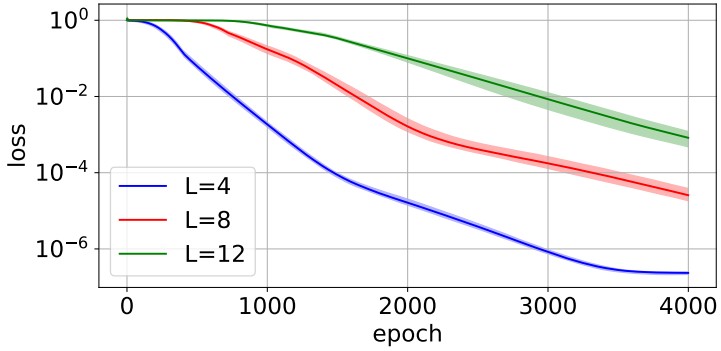

Figure 4: Training loss curves for networks of depth $L = 4$, $L = 8$, and $L = 12$.

