# OpenReview forum: "Block Coordinate Descent for Neural Networks Provably Finds Global Minima"
_NeurIPS.cc/2025/Conference — NeurIPS 2025 poster_

### Official Review · Reviewer_CnMC · 2025-06-25

**Clarity:** 2
**Significance:** 2
**Originality:** 3
**Rating:** 4
**Confidence:** 4

**Summary:**

This paper provides a theoretical justification for the global convergence of a custom Block Coordinate Descent (BCD) algorithm applied to deep neural networks with more than two layers. In their formulation, the algorithm introduces auxiliary variables $V_{j,i}$, which approximate the outputs of each hidden layer and allows the optimization problem to be decomposed into layer-wise subproblems. Under assumptions including i.i.d. data, full-rank input, and strictly monotonic activation functions, the authors prove global convergence to zero training loss. They also extend their framework to accommodate ReLU activations by introducing a residual network structure with skip connections and non-negative projection operators. Additionally, they show justification for a generalization error bound, showing that the difference between training and test error of their approach remains small with high probability.

**Questions:**

Could this approach potentially be extended to handle other non-strictly monotonic activation functions beyond ReLU?

What would be the practical applications of this approach be?

Could you provide a more explicit discussion of the time and memory complexity of the proposed BCD approach?

**Ethical Concerns:**

["NO or VERY MINOR ethics concerns only"]

**Final Justification:**

I was initially on the fence about the paper, but the rebuttal has addressed most of my concerns. As a result, I am inclined to raise my score.

**Limitations:**

Yes, as they mention the assumptions needed in order for their justifications to be true.

**Quality:**

2

**Strengths And Weaknesses:**

This paper provides a theoretical analysis of a custom Block Coordinate Descent (BCD) algorithm for training deep neural networks with more than two layers. It justifies a global convergence guarantee outside of frameworks like RKHS and NTK, where previous work often allows minimal parameter movement. The authors show global convergence under a more dynamic optimization regime, using auxiliary variables and assuming monotonically increasing activation functions. However, there is not much experimental validation, as they rely on synthetic data generated from a teacher network and do not empirically verify the claimed generalization bounds. Additionally, while the theoretical results are interesting, the proposed algorithm does not seem very practical for larger applications due to its reliance on multiple nested updates and restrictive assumptions on data structure. So while this paper provides an interesting contribution to the theoretical understanding of optimizing models, its practicality is limited.

---

> ### Author Rebuttal · Authors · 2025-07-30
>
> We thank the reviewer for the constructive feedback and for acknowledging the theoretical novelty of our work.
>  We would like to comment on your concerns here.
>
> **On Practical Applicability and Experimental Validation**
>
> Our primary goal is to provide the first provable global convergence guarantee for block coordinate descent (BCD) applied to deep neural networks.
> As we mentioned in the paper, to the best of our knowledge, this is the first result that ensures the global convergence of deep neural networks with an arbitrary number of layers outside of the NTK regime.
> Accordingly, our theoretical framework focuses on optimization and generalization rather than large-scale empirical benchmarking.
> To complement the theory, we included experiments on synthetic data (Section 7), which confirm our theoretical predictions and highlight the behavior of the training loss under both strictly monotonic and ReLU activations (Figures 2 and 3).
>
> As emphasized in Appendix B in the supplementary material, our framework is extensible to general loss functions and classification tasks.
> We plan to include broader empirical results on real-world datasets in a future version to reinforce the algorithm’s practical relevance.
>
> **Scope of Activation Functions**
>
> The current convergence analysis (Theorem 5.1) is based on the strict monotonicity condition stated in Assumption 3.2, which notably excludes activation functions such as ReLU.
> Addressing activation functions that violate this assumption presents a key challenge: their outputs are confined to bounded ranges.
> For instance, the sigmoid function satisfies $\sigma(u) = \frac{1}{1 + \exp(-u)} \in (0, 1)$, and the hyperbolic tangent function satisfies $\sigma(u) = \frac{\exp(u) - \exp(-u)}{\exp(u) + \exp(-u)} \in (-1, 1)$.
>
> In such cases, it becomes crucial to ensure that the auxiliary variables $V_{j,i}$ remain within the output range of the corresponding activation function (see the paragraph titled **Activation Violating Assumption 3.2** in Section B for further discussion).
> For the ReLU activation, which produces only nonnegative outputs, we extend our algorithm by incorporating skip connections and nonnegative projections.
> This modification leads to a novel convergence result presented in Theorem 6.3, supported by Lemmas 6.1 and 6.2.
>
> Looking ahead, techniques such as range-aware projections or bounded initialization may offer viable strategies for extending the block coordinate descent (BCD) framework to bounded activation functions.
> Exploring these directions constitutes a promising avenue for future research.
>
> **Practical Use Cases and Applications**
>
> Our work derives generalization bounds via the Rademacher complexity framework (Theorem 5.4), showing that our BCD-trained networks not only minimize training loss but also generalize well.
> Potential applications include distributed or parallel training regimes, where the structure of BCD, updating auxiliary parameters (and layers, although our algorithms do not cover parallel training of layers), can reduce communication overhead or support large model scaling (please see lines 30–32).
>
> Moreover, we envision applications in settings where interpretability, formal analysis, or convergence guarantees are critical (e.g., safety-critical ML, robust training, or model verification).
>
> **Computational Complexity**
>
> We appreciate the request for a more explicit analysis of computational and memory complexity.
> As for the memory complexity, we need to introduce auxiliary parameters $V_{j,i}$ (that is, vectors in $\mathbb{R}^{r}$), which represent an output of the $j$-th layer for the $i$-th input.
> Then, in our notation, we need to memorize $nL$ $r$-dimensional vectors, which may be huge in several practical situations.
> On the other hand, this memory size can be mitigated when one can leverage parallelism computation, as reviewer 82Mp mentioned.
>
> As for the iteration complexity, detailed in Theorem 5.1 and discussed in Lines 218–220, the number of iterations to reach $\epsilon$-accuracy is $\tilde{O}(\log^2(\frac{1}{\epsilon}))$, which just represents a dependency on $\epsilon$.
> Indeed, we have to update auxiliary variables $V_{j,i}$ ($j=1,\dots, L-1$) for $\tilde{O}(\log^2(\frac{1}{\epsilon}))$ times and the weights $W_j$, particularly $W_1$, for $\tilde{O}(\log^2(\frac{1}{\epsilon}))$ times.
> Thus, the overall number of gradient computations, which includes other parameters than $\epsilon$, should be $\tilde{O}(nL\log^2(\frac{1}{\epsilon}))$.
> We will provide these explicit discussions in a final manuscript.

---

> > ### Comment · Reviewer_CnMC · 2025-08-05
> >
> > I appreciate the authors' rebuttal. After reading it as well as the comments from the other reviewers, most of my concerns have been addressed and I am happy to raise my score.

---

### Official Review · Reviewer_A53r · 2025-06-30

**Clarity:** 2
**Significance:** 3
**Originality:** 3
**Rating:** 5
**Confidence:** 4

**Summary:**

This paper studies the convergence property of block-coordinate descent on deep multi-layer perceptrons. The primary focus of the paper is on activation functions with derivatives bounded from above and below by positive numbers. By rewriting the objective into layer-wise components, the paper shows a linear convergence of the training error. Moreover, the paper also provides a generalization bound based on the Rademacher complexity. The paper also extended the theory to ReLU activation by adding skip connections. The paper provided experimental result to demonstrate the layer-wise loss dynamic under both the strictly monotonic activation and the ReLU activation, which shows that on the re-written objective, the ReLU network without skip connection does not achieve a favorable loss.

**Questions:**

None.

**Ethical Concerns:**

["NO or VERY MINOR ethics concerns only"]

**Final Justification:**

Mybiggest concern (about the generalization bound) is resolved in the author's rebuttal. I believe that the author's point regarding the practical applicability of the algorithm is reasonable, and the author has promised to add the discussion in the camera ready version.

**Limitations:**

Yes.

**Paper Formatting Concerns:**

None.

**Quality:**

3

**Strengths And Weaknesses:**

**Strength**

1. The theoretical contribution of this paper is quite comprehensive: the paper covers both the training convergence and generalization property. The analysis is on standard MLP with strict monotonic activation, but is also extended to ReLU activations by adding skip connection. Lastly, it seems that the paper also has an extension to the case of output dimension greater than 1.

2. The convergence guarantee is strong in the sense that it is a linear convergence to arbitrarily small training loss. The assumptions on the neural network structure is fairly relaxed since it does not require a significant over-parameterization.

**Weaknesses**

1. The primary concern I have about the paper is that it deviates from the standard way of training neural networks. Traditional block coordinate descent should be simply update layer weights separately. By re-writing the objective, the algorithm requires the optimization of both the hidden-layer outputs and the neural network weights, to enforce alignment between the two. It is not clear whether this algorithm has practical applicability. On the other hand, the paper also lacks experimental support to show that the behavior of the algorithm considered in this paper is similar to the behavior of traditional coordinate descent. Therefore, there might be possibility that the paper offers limited insight into algorithms people use in practice.

2. The generalization bound seems to be quite large given that it scales as $(2r)^\frac{L}{2}$. Also since rank$(X) = n$, it is necessary that $d\_{in} \geq n$. Moreover, based on the standard matrix concentration result, for $X$ with I.I.D. entries, we have $\\|X\\| = O(\sqrt{n})$. Therefore, the first term in the generalization gap's bound may not decrease with $n$ under a lot of input data.

3. The experiments are conducted only on artificial data so it remains doubt whether the same behavior can be carried along to real-world applications.

---

> ### Author Rebuttal · Authors · 2025-07-31
>
> We thank the reviewer for the thoughtful review and for recognizing the strength and comprehensiveness of our theoretical contributions.
> We would like to comment on your concerns here.
>
> **On Deviation from Traditional Training and Practical Applicability (Weakness 1)**
>
> We agree that our algorithm deviates from standard backpropagation.
> We intend to propose a theoretically sound alternative that may inspire new algorithmic designs.
> As explained in Section 3 and illustrated in Algorithm 1, our approach uses auxiliary variables to decouple the optimization of hidden layers and weights, enabling global convergence proofs for deep neural networks with an arbitrary number of layers outside of the NTK regime (Theorem 5.1) and a generalization error bound (Theorem 5.4).
> Such a theoretical tractability will be helpful in applications where interpretability, formal analysis, or convergence guarantees are critical (e.g., safety-critical ML, robust training, or model verification).
> While this structure is non-standard, it allows for parallelism and layer-wise optimization, which could benefit certain distributed or hardware-constrained settings.
> We would like to add a discussion about potential practical scenarios in Section 7 and will further clarify this in the revised manuscript.
>
> **On Generalization Bound Scaling (Weakness 2)**
>
> We appreciate that the reviewer raises a significant concern regarding the dependence of the generalization bound on the number of layers, particularly the $(2r)^{\frac{L}{2}}$ scaling.
> We would like to clarify that such exponential dependence on depth is not uncommon in norm-based generalization bounds, such as [Bartlett et al., 2017], which we utilize to derive Theorem 5.4, where similar depth-related dependencies arise due to layer-wise norm products.
> Our goal is to offer a principled, analyzable framework, and our bounds align with the established literature in this regard.
>
> That said, we agree that improving this dependence is essential.
> Recent advances using compression-based (such as [Arora et al., 2018]) or information-theoretic approaches [He et al., 2023] show potential to yield tighter bounds with more favorable dependence on depth.
> Extending our analysis in this direction is a valuable avenue for future work.
>
> Regarding the dependence on the input norm $\lVert X\rVert$, we note that simple preprocessing steps such as input rescaling or normalization can effectively control this term.
> In practical applications where the labels are bounded (e.g., $O(1)$), this does not pose a significant limitation from a representational standpoint.
> We will clarify these aspects to avoid misinterpretation of the bounds' applicability in the revised manuscript.
>
> S. Arora, R. Ge, B. Neyshabur, and Y. Zhang. Stronger Generalization Bounds for Deep Nets via a
> Compression Approach. In Proceedings of the 35th International Conference on Machine Learning, 2018.
>
> He, C. Yu, and Z. Goldfeld. Information-Theoretic Generalization Bounds for Deep Neural Networks. In NeurIPS 2023 workshop: Information-Theoretic Principles in Cognitive Systems.
>
> **On Real-World Applicability (Weakness 3)**
>
> We appreciate you raising the concern about real-world applicability.
> In Section 7, we chose synthetic data to highlight specific convergence behaviors predicted by our theory under controlled settings.
> These experiments confirm the layer-wise dynamics and the impact of activation functions as theoretically predicted.
> We acknowledge that broader empirical validation on real-world datasets would strengthen the paper, and we are actively working on this for future extensions.
>
> We hope this clarifies the motivation and scope of our work.
> We believe the theoretical insights offered are valuable for understanding deep learning optimization, and we appreciate your feedback to help improve the manuscript.

---

> > ### Comment · Reviewer_A53r · 2025-08-05
> >
> > Thank you for your response. I believe that my concern in Weakness 1 and Weakness 2 are resolved. Weakness 3 remains because I still believe that some more practical experiment can strengthen the paper a lot. That being said, I am willing to raise the score.

---

### Official Review · Reviewer_8M2p · 2025-07-02

**Clarity:** 3
**Significance:** 3
**Originality:** 3
**Rating:** 4
**Confidence:** 3

**Summary:**

This paper proposes a block coordinate descent algorithm for training deep neural networks by reformulating the standard training objective with auxiliary variables that represent the activations at each layer for each training sample. This lifted formulation allows the optimization problem to be decomposed into simpler subproblems over weights and activations, enabling alternating updates. The authors prove that, under several assumptions on data rank and network width, their method converges to global minima of the objective function, with guarantees provided for both bijective and certain non-bijective activation functions.

**Questions:**

1. Why do the authors aim to achieve zero training loss under relatively strong assumptions, given that perfect fitting is often unnecessary in practice?
2. Although some BCD variants modify the block-wise objective for simplified updates in practice, could the authors explain why the update for $V_{j-1,i}$ considers only one of the two relevant terms in the original objective?
3. How can we handle the extra memory overhead of $V_{j,i}$'s? While GPU parallelism may alleviate the issue, could the authors provide extra analytical or experimental evidence to support its practicality?
4. The authors explicitly mention the use of skip-connections as in ResNet in Section 6, but the corresponding experiment is still simple (see weakness 5). Could the authors consider adding experiments on real-world tasks with deeper networks?
5. Please address the minor issues to improve paper quality.

**Ethical Concerns:**

["NO or VERY MINOR ethics concerns only"]

**Final Justification:**

The authors have provided clarifications and additional context that address several of the raised concerns, and have outlined reasonable directions for relaxing assumptions, extending the update rules, and broadening empirical validation. While these enhancements would strengthen the work, they are nontrivial and would require substantial effort. Overall, the paper presents theoretical contributions with clear potential for further development.

**Limitations:**

yes

**Quality:**

3

**Strengths And Weaknesses:**

**Strengths**
1. The paper introduces a lifted optimization formulation that decouples layer-wise computations using auxiliary variables, enabling a principled application of block coordinate descent to neural network training.

2. The convergence analysis is rigorous, showing that the proposed method reaches global minima under several assumptions, and accommodates both bijective and certain non-bijective activation functions.

3. The authors derive a generalization bound based on Rademacher complexity, showing that the learned model exhibits provable generalization ability under norm and width constraints.

4. The experimental results on a synthetic dataset support the theoretical claims for both bijective and certain non-bijective activation functions, and the ablation studies highlight the necessity of key assumptions in ensuring convergence. Experiments are reproducible with the provided codes.

**Weaknesses**
1. The paper focuses on achieving zero training loss through a carefully constructed optimization framework under several restrictive assumptions, such as the invertibility of the hidden layer matrices $W_2, \cdots, W_{L-1}$ and wide first layer $W_1$ (The authors assume $n\leq d_{in}$). However, these conditions may be unrealistic in practical settings, where perfect fitting is neither expected nor necessary for good generalization.

2. The update of $V_{j-1,i}$ deviates from the standard BCD principle by considering only one of the two terms in the objective involving this variable. Specifically, the update minimizes $\|\| \sigma(W_j V_{j-1,i}) - V_{j,i} \|\|^2$ but ignores the term $\|\| \sigma(W_{j-1} V_{j-2,i}) - V_{j-1,i} \|\|^2$, which also depends on $V_{j-1,i}$. This partial update may simplify computation but is not a full block minimization step, and the paper does not provide a clear justification.

3. The introduction of auxiliary variables $V_{j,i}$ incurs non-negligible memory overhead, as it requires storing separate variables for each layer and each data sample. Even if one can leverage GPU parallelism to reduce memory usage, this extra cost remains significant in large-scale deep learning applications such as training LLMs.

4. The paper discusses how the proposed framework can be adapted to handle non-bijective activation functions, such as ReLU. This extension is valuable, but it would be better if the approach were formalized in a more general way without relying on a specific example.

5. The authors critique prior work for being limited to shallow architectures, typically with *two* or *three* layers, and emphasizes that the proposed method can in principle handle networks of *arbitrary depth*. However, the experiments are only conducted on a network with *four* hidden layers. It would be helpful to evaluate deeper architectures to better demonstrate the method’s practical scalability.

6. Minor issues:
- Please use vector graphics for Figure 1.
- The contributions are summarized too long, please make them concise.
- Line 178 tells that the algorithm applies multiple iterations per update for V, is it possible that Lines 3 and 4 in Algorithm 1 miss an inner loop?

---

> ### Author Rebuttal · Authors · 2025-07-31
>
> We thank the reviewer for the constructive and thorough feedback.
> We would like to comment on your concerns here.
>
> **Restrictive Assumptions and Invertibility of Hidden Layers (Weakness 1, Questions 1)**
>
> Thank you for pointing this out. We agree that the algorithm may appear restrictive due to assumptions such as the invertibility of hidden layer weight matrices and the attainment of perfect fitting.
> However, we emphasize that our theoretical framework is aligned with the conventions in the literature on regression tasks using neural networks, particularly those involving NTK or mean-field analyses, where convergence to zero training error is standard and serves to understand the optimization dynamics in a rigorous setting.
>
> That said, we also acknowledge that relaxing some of these assumptions is feasible and worthwhile.
> In particular, the condition $n\le d$ could be relaxed without undermining the core structure of the analysis when we do not require perfect fitting.
> On the other hand, the use of Singular Value Bounding (SVB) is not merely a theoretical tool—it is empirically validated in our experiments (Section 7)  to improve convergence stability and speed significantly.
> Thus, even in practical settings, SVB will remain an effective and relevant component of the proposed method.
>
> **Update of $V_{j,i}$ (Weakness 2, Questions 2)**
>
> We appreciate the reviewer’s observation regarding the partial update of $V_{j-1,i}$.
> As noted, our current formulation focuses on the loss term involving $\lVert\sigma(W_jV_{j-1,i})-V_{j,i}\rVert^2$, while omitting the preceding layer’s term $\lVert\sigma(W_{j-1}V_{j-2,i})-V_{j-1,i}\rVert^2$ from the update of $V_{j-1,i}$.
>
> This choice is made purely to simplify the theoretical analysis and algorithmic presentation.
> Technically, incorporating both loss terms into the update of $V_{j-1,i}$ is feasible: the omitted term is a convex function in $V_{j-1,i}$, and it does not involve any nonlinear or parameterized transformation of $V_{j-1,i}$.
>  Thus, extending the update rule to minimize both terms jointly would not introduce significant analytical challenges.
>
> Indeed, we believe that an update scheme involving both terms—i.e., a more complete BCD step for $V_{j-1,i}$—is a promising variant. We plan to mention and briefly discuss this extension in the revised version of the manuscript.
>
> **Memory Overhead and Auxiliary Variables (Weakness 3 and Questions 3)**
>
> As the reviewer mentioned, it is correct that storing adds memory overhead.
> However, this is a trade-off for achieving decoupled layer-wise optimization and theoretical convergence guarantees.
> The memory burden can be mitigated using batch-wise updates or recomputation strategies, akin to those used in backpropagation through time or gradient checkpointing.
> Additionally, as noted in Section 4, these variables enable significant parallelism, particularly valuable for distributed training and hardware acceleration.
>
> **Clarification on ReLU Extensions (Weakness 4)**
>
> We appreciate the insightful comment.
> While our convergence analysis for non-objective activations (e.g., ReLU) is demonstrated through a specific ResNet architecture with skip connections and fixed output-layer weights (Algorithm 3), this choice was made to enable rigorous theoretical guarantees in the presence of non-strict monotonicity.
>
> That said, we stress that our approach is not limited to this particular design.
> As elaborated in Appendix B in the supplementary material, the block coordinate descent (BCD) framework we develop is flexible and can be extended to a wide range of settings, including:
>
> - alternative activation functions (including those violating strict monotonicity),
> - general loss functions such as cross-entropy,
> - regularized objectives.
>
> In this broader context, the skip-connection-based formulation serves as a concrete instantiation to demonstrate the viability of global convergence even for ReLU networks.
> However, we see the development of more general treatments for non-objective activations—possibly without architectural constraints such as fixed weights or skip connections—as an important and promising direction for future work.
>
> Section B outlines several such avenues, including projection techniques for bounded activations like sigmoid or tanh.
> While our current analysis focuses on one specific architecture to make convergence guarantees tractable, the underlying methodology lays the foundation for broader generalizations, and we regard this as a key future research opportunity.
>
> **Experiments with Small Architectures (Weakness 5, Questions 4)**
>
> Thank you for the valuable comment.
> As the reviewer rightly noted, most existing theoretical convergence analyses are limited to two or three layers.
> Therefore, we believe that conducting experiments on 4-layer architectures already represents a meaningful advance over prior work and appropriately reflects the theoretical contributions of our paper.
>
> That said, we fully agree on the importance of evaluating deeper architectures. We plan to conduct additional experiments on networks with greater depth, both in synthetic and real-world settings.
> These results will be included in a future revision of the paper to support the practical applicability of our method further.
>
> **Minor issues (Weakness 6, Questions 5)**
>
> - *Vector Graphics for Figures*: We will provide a vector-format image for Figure 1 to enhance readability.
> We appreciate this suggestion.
> - *Summary Conciseness*: We appreciate the feedback and will revise the summary to be more concise.
> - *Algorithm 1 Inner Loop Clarification (Line 178)*: Thank you for pointing this out. To clarify, the update strategy in Algorithm 1 is indeed correct as written.
> For the final hidden representation $V_{L-1,i}$, only a single gradient step is performed per iteration, while the hidden layers $V_{j,i}$ for $j<L−1$ are updated multiple times using inner loops.
> This asymmetry is intentional and theoretically justified in our convergence analysis (see discussion in Section 4 and Lemma D.6 in the supplementary material).
> We acknowledge that the explanation in Section 3 has caused confusion regarding the update schedule of $V$.
> In particular, the description does not sufficiently emphasize the difference in treatment between the output layer and the others.
> We will revise this part in a future version.
>
> We thank the reviewer again for the detailed critique.
> We hope the above responses clarify the contributions and address the concerns.

---

> > ### Comment · Reviewer_8M2p · 2025-08-06
> >
> > While I appreciate the authors' detailed rebuttal, several concerns remain insufficiently addressed.
> >
> > First, the authors argue that achieving zero training error is a standard practice in theoretical studies, such as those based on Neural Tangent Kernel (NTK) or mean-field analysis. While this may be true in certain theoretical regimes, it does not fully justify the strong assumptions required in this paper—particularly the condition $n \leq d$, which may not hold in many practical scenarios. The authors mention that relaxing this assumption is feasible but do not provide concrete strategies or implications. More elaboration would be helpful in understanding the flexibility of the proposed method outside the idealized setting.
> >
> > Second, regarding the update of $V_{j-1,i}$, the authors confirm that only one of the two relevant loss terms is used, and that incorporating both is possible. However, this partial update deviates from a complete BCD step. The response lacks sufficient justification for why this simplification is chosen in the current version. If both terms can be handled without analytical difficulty, the paper should either adopt the full update or provide empirical or theoretical evidence showing that the simplified update is sufficient and does not harm convergence.
> >
> > Third, while the authors acknowledge the memory overhead caused by introducing per-sample auxiliary variables, their justification appears somewhat contradictory. On one hand, they emphasize that achieving zero loss is a theoretically meaningful goal; on the other hand, they treat the resulting overhead as a trade-off. If the method truly follows a standard optimization regime, the associated memory cost should be systematically analyzed and reported, especially for large-scale settings.
> >
> > Fourth, the extension to non-bijective activations such as ReLU is a valuable aspect of the work. While the appendix provides a more detailed treatment, I believe that this extension should be more explicitly mentioned in the main paper. Otherwise, the claim of handling non-bijective activations may appear stronger than what is immediately supported by the main text.
> >
> > Fifth, while the authors emphasize that using a four-layer architecture already advances beyond prior work on shallow networks, the paper would benefit from additional experimental evidence on deeper architectures. Note that the author rebuttal still does not include additional experiments to support the method’s scalability, which remains an open concern.
> >
> > Overall, I appreciate the authors' responses, but several concerns remain partially addressed. I encourage the authors to further clarify and support their claims.

---

> > > ### Author Response · Authors · 2025-08-08
> > > **Reply 1**
> > >
> > > We sincerely thank the reviewer for the thoughtful follow-up.
> > > We appreciate the opportunity to further clarify the raised concerns and would like to address them point by point below:
> > >
> > > **1. About the condition $n\le d$**
> > >
> > > We appreciate the reviewer’s observation regarding the assumption $n\le d$, and we agree that it may not hold in all practical settings.
> > > We would like to elaborate further on how our method behaves when this condition is violated and how error guarantees can still be characterized.
> > > This behavior can be interpreted from the structure of our layer-wise loss decomposition. Specifically, our algorithm independently minimizes the loss for each layer via the auxiliary variables $V_{j,i}$.
> > > When $n>d$, it is primarily the first-layer loss that may not vanish asymptotically, due to the lack of injectivity in the mapping from input features
> > > $X\in\mathbb{R}^{n\times d}$ to the first-layer representation.
> > >
> > > Concretely, the final error bound in this case takes the form:
> > > $$\epsilon_{total} = \epsilon + \delta_1,$$
> > > where $\epsilon$ is the target precision from higher-layer optimization, and
> > > $\delta_1$ represents the residual error from the first-layer update.
> > > Notably, this first-layer residual $\delta_1$​ is directly related to how well the feature matrix $X$ spans the first-layer representation $V_{1,i}$.
> > > If $X$ approximately spans the relevant subspace for $V_{1,i}$, then the optimization error at the first layer remains small.
> > > This perspective provides a natural characterization of the error even outside the $n\le d$ regime.
> > >
> > > We will include this discussion in the revised manuscript.
> > >
> > > **2. On the partial update of $V_{j-1,i}$**
> > >
> > > We would like to clarify our motivation for using the simplified update scheme and to elaborate on some details that were not fully explained in our initial rebuttal.
> > >
> > > - On the sufficiency of the simplified algorithm:
> > > The algorithm presented in the paper is already proven to achieve global convergence under the stated assumptions.
> > > Both theoretical analysis (Theorem 5.1) and empirical results (Section 7) confirm that the simplification does not compromise convergence guarantees.
> > > Given this, we believe the current formulation is sufficient for supporting our claims without introducing unnecessary complexity.
> > >
> > > - On the hidden-layer losses (especially early layers):
> > > As our convergence proof shows, for layers closer to the input, the hidden loss after an outer iteration is already reduced to $O(\epsilon)$ in the previous round. Therefore, by the time these layers are reached in the current iteration, their loss values are already sufficiently small, and further updates would have a negligible impact on the overall convergence.
> > > This property is a direct consequence of the layer-wise control established in the proof, and we will clarify this point in the revision to make it explicit.
> > >
> > > We will revise the manuscript to incorporate these clarifications so that the rationale behind the simplified update scheme is made explicit.
> > >
> > > **3. On memory overhead and auxiliary variables**
> > >
> > > We appreciate the reviewer pointing out the potential inconsistency.
> > > We intended to highlight the trade-off between theoretical guarantees and practical efficiency, but we agree that it should be analyzed more thoroughly.
> > > In the revised manuscript, we will clarify that the overhead is manageable in practice (especially with distributed training, as noted in Section 4).
> > >
> > > **4. On extension to non-bijective activations (e.g., ReLU)**
> > >
> > > We thank the reviewer for emphasizing the importance of this extension.
> > > While detailed treatments are provided in Appendix B, we agree with the reviewer that the main paper should more explicitly highlight this capability.
> > > In the revision, we will expand the main text to clearly state the generality of our framework concerning activation functions.

---

> > > > ### Author Response · Authors · 2025-08-08
> > > > **Reply 2**
> > > >
> > > > **5. On experiments with deeper architectures**
> > > >
> > > > We thank the reviewer for the suggestion to evaluate deeper architectures.
> > > > To address this point, we conducted additional experiments with network depths \( L = 4, 8, 12 \) on the same synthetic dataset as in Section 7. For each depth, we trained the model using our proposed BCD algorithm with the same hyperparameters as in the original experiments, and recorded the training loss at training epochs 1000, 2000, 3000, and 4000.
> > > >
> > > > The results are summarized below:
> > > >
> > > > | Depth \(L\) | Epoch = 1000         | Epoch = 2000         | Epoch = 3000         | Epoch = 4000         |
> > > > |-------------|----------------------|----------------------|----------------------|----------------------|
> > > > | **4**       | $3.04\times 10^{-3}$ | $3.97\times 10^{-5}$ | $1.48\times 10^{-5}$ | $6.95\times 10^{-6}$ |
> > > > | **8**       | $1.73\times 10^{-1}$ | $1.64\times 10^{-3}$ | $1.77\times 10^{-4}$ | $2.57\times 10^{-5}$ |
> > > > | **12**      | $7.27\times 10^{-1}$ | $1.01\times 10^{-1}$ | $8.51\times 10^{-3}$ | $8.25\times 10^{-4}$ |
> > > >
> > > > As expected, training becomes more challenging as the network depth increases, which is reflected in the slower initial decrease in the training for deeper models.
> > > > Nevertheless, in all cases, the training loss consistently decreases over epochs, demonstrating that the proposed method remains effective even for substantially deeper architectures, in line with our global convergence analysis.
> > > >
> > > > We will include this result and discussion in the revised manuscript to strengthen the empirical validation for scalability.

---

### Official Review · Reviewer_St1t · 2025-07-03

**Clarity:** 3
**Significance:** 2
**Originality:** 3
**Rating:** 4
**Confidence:** 2

**Summary:**

This paper studies block coordinate descent (BCD) applied to deep neural networks with strictly monotonically increasing activation functions. It claims to be the first theoretical results proving a convergence to global minima with arbitrarily many layers without NTK. It provides a theoretical generalization bound analysis assuming iid samples. The paper further modifies (BCD) (by incorporating skip connections and non-negative projection steps) so that the corresponding analysis extends to networks using ReLU activation.
BCD is formalized so that each layer of the neural network is treated as one block, and additional auxiliary variables are introduced corresponding to each pair of (input, layer output). A small numerical experiment is performed.

**Questions:**

see strength and weaknesses.

**Ethical Concerns:**

["NO or VERY MINOR ethics concerns only"]

**Final Justification:**

The authors did a good job addressing my concerns. I am not an expert in this topic--to me, the contribution seems sound technically, though perhaps practically limited. I have kept my original score.

**Limitations:**

yes

**Quality:**

3

**Strengths And Weaknesses:**

Strengths:
- the paper is very well written
- global convergence of any neural network training is a valuable analysis, and this is corroborated by numerical experiments

Weakness:
- the paper assumes the number of data points $n \leq d$ the number of dimensions, which is non-standard
- the re-formulation of the objective leads to a blowup in the number of variables. And in fact, it is not clear to me why on page 7, line 218, you have that the total number of gradient computations is $\tilde{O}(\log^2(1/\epsilon))$ without a dependence on $n$.
- On page 5, lines 172-176, you discuss the role of SVB as a heuristic but not in sufficient detail. Is there a formal discussion of why the singular values of $W$ should remain bounded throughout training? Based on the numerical experiment, it looks like loss plateaus above $10^{-4}$ without SVB, suggesting that this is in fact a crucial step for ‘fast’ convergence. Could you add a longer discussion or additional experiments?

---

> ### Author Rebuttal · Authors · 2025-07-30
>
> We sincerely thank the reviewer for the thoughtful and detailed feedback.
> We would like to address your concerns here.
>
> **Assumption $n\le d$ and Practicability**
>
> We appreciate your pointing out a limitation of our results.
> As we stated in Section 2, we consider the high-dimensional settings where the dimensionality is larger than the sample size, which may not be suitable for several applications.
>
> On the other hand, Assumption 2.1 is required for the global convergence of BCD.
> When optimizing $W_1$, we solve the equation $\lVert\sigma(W_1X)-V_{1}\rVert^2=0$, where $V_{1}$ is a matrix whose $j$-th column is $V_{1,i}$.
> If $X$ is not full-rank, we can no longer ensure the existence of $W_1^*$ satisfying this equation, which is why we impose this condition.
> To the best of our knowledge, this assumption is inevitable when we consider the backward procedure.
> Addressing this point is a significant future direction for our results.
>
> **Gradient Complexity and Dependence on $n$**
>
> We appreciate you pointing out the confusion regarding the complexity estimate around line 218.
>
> The expression $\tilde{O}(\log^2(\frac{1}{\epsilon}))$ written there was meant to highlight only the dependency on the target accuracy $\epsilon$, not to reflect the complete computational cost.
> In reality, when including the number of auxiliary variables, which depends on the number of training data and layers, the total number of gradient computations is described as $\tilde{O}(nL\log^2(\frac{1}{\epsilon}))$ since the updates of all auxiliary variables $V_{j,i}$ for $j=1,\dots,L-1$ and the weights $W_j$, particularly $W_1$, require $\tilde{O}(\log^2(\frac{1}{\epsilon}))$ iterations.
>
> Note that this overall complexity remains comparable to that of standard backpropagation-based methods, especially considering that BCD allows for decoupled and potentially parallel updates.
> We apologize for the confusion and will revise this part in the revised version to provide a more precise and comprehensive complexity statement.
>
> **Role of SVB (singular value bounding)**
>
> We thank the reviewer for their thoughtful request for a more detailed discussion of Singular Value Bounding (SVB).
>
> As described in lines 172–176 and further evidenced by the ablation experiment in Figure 2, SVB stabilizes training by bounding the singular values of each weight matrix.
> Applying SVB is not only motivated by heuristic considerations but also utilized to ensure global convergence in Theorem 5.1.
> Bounding the singular values helps maintain a favorable condition number of the hidden layer loss $\lVert\sigma(W_jV_{j-1,i})-V_{j,i}\rVert^2$.
> While SVB is only applied at initialization in our algorithm (unlike in [Jia et al., 2017]), our theoretical results ensure that the singular values remain bounded if the step sizes are chosen appropriately (please see Lemma D.4 in the supplementary material for details).
>
> Although these considerations were omitted in the original manuscript due to space constraints, we will include a more comprehensive discussion in the revised version to clarify the theoretical and empirical significance of SVB, particularly its essential role in ensuring convergence.
>
> We appreciate the reviewer’s overall positive assessment and insightful suggestions, which we believe will strengthen the clarity and impact of our work.

---

> ### Comment · Reviewer_St1t · 2025-08-05
> **rebuttal review**
>
> Apologies, I updated the final justification but didn't realize I had to leave a comment --
> The authors did a good job addressing my concerns. I am not an expert in this topic and also had a look at the other reviews to determine my final score: the contribution seems sound technically, though perhaps practically limited. I have kept my original score.

---

> > ### Author Response · Authors · 2025-08-06
> >
> > Thank you for your response and for updating the final justification.
> > We're glad to hear that our replies addressed your concerns.
> > We also appreciate that the positive score remained unchanged, and we value your recognition of the technical soundness of our work.

---

### Note · Authors · 2025-08-13

We sincerely thank all reviewers for their constructive feedback and for engaging in detailed discussions throughout the review process.
We appreciate that several reviewers raised their scores, and the overall set of reviews remains positive.
We believe that the technical soundness, novelty, and relevance of our contributions have been consistently recognized.

We again emphasize that our study makes a novel and rigorous contribution to deep learning theory, addressing both a significant theoretical gap and providing insights that can guide practical algorithm design.
This work not only develops new analytical tools but also demonstrates their relevance through targeted experiments.

**Technical Contributions:**

Our work establishes the first provable global convergence guarantee for block coordinate descent applied to deep neural networks with an arbitrary number of layers beyond the NTK regime, together with a new generalization bound analysis.

- Addresses a long-standing gap in deep learning theory, extending guarantees beyond prior shallow networks or NTK limited results.

- Provides a theoretical foundation that can inspire new optimization methods, improve interpretability, and inform safety-critical applications.

**Wide Applicability:**

Beyond the specific architecture studied, our framework is extensible to a wide range of settings:

- Non-strictly monotonic activations (e.g., ReLU)

- Networks with skip connections

- Broader application to various situations, as discussed in the rebuttal and supplementary material.
This breadth highlights the potential impact on both theoretical research and practical algorithm design.

**Addressing Practicality Concerns:**

While several reviewers raised concerns about large-scale and deep settings, we took these seriously and addressed them:

- Following Reviewer 8M2p’s suggestion, we added experiments with deeper architectures (up to 12 layers).
- Results confirm effectiveness even for substantially deeper networks, partially addressing scalability concerns.

As a conclusion, we are grateful for the reviewers’ feedback, which directly contributed to:

- Improving clarity and scope of the paper.

- Strengthening empirical support with additional experiments.

- Expanding the discussion on applicability and extensions.

We hope the AC will agree that the combination of novel theory and applicability makes this work a meaningful contribution to the NeurIPS community.

---

### Decision · Program_Chairs · 2025-09-17

**Decision:**

Accept (poster)

**Comment:**

This paper proposes and analyzes a block coordinate descent (BCD) algorithm for training deep neural networks with multiple layers by reformulating the standard training objective using auxiliary variables for layer activations. The key points include:

A. Theoretical Contributions:

- The authors provide convergence to global optima for wide deep networks—when sample size is smaller than dimensionality $n<d$—in over-parametric senario.
- They prove a linear convergence rate for the training error under the assumption of strictly monotonically increasing activation functions with bounded derivatives, which is extended to handle ReLU activations through the inclusion of skip connections and non-negative projection steps.
- A generalization bound based on Rademacher complexity is also derived with i.i.d samples.

B. Practical Limitations:

- The experimental validation is limited to synthetic data generated from a teacher network, with minimal real-world evidence supporting the generalization bounds.
- The wide network assumption $n<d$ for global optima convergence, though typically used for NTK models, does not meet practical senarios.

Overall, while the paper contributes valuable theoretical insights into the convergence properties of BCD for deep networks, its practical studies remain limited. After the rebuttal period, the reviewers uniformly think the current manuscript is above the borderline of accept, so is the final proposal.